# *Cryptosporidium* uses CSpV1 to activate host type I interferon and attenuate antiparasitic defenses

Silu Deng[1,2], Wei He[2], Ai-Yu Gong[1,2], Min Li[2], Yang Wang[2], Zijie Xia [2], Xin-Tiang Zhang[2], Andrew S. Huang Pacheco[3], Ankur Naqib[4], Mark Jenkins[5], Patrick C. Swanson[2], Kristen M. Drescher[2], Juliane K. Strauss-Soukup [6], Michael Belshan[2] & Xian-Ming Chen [1,2] ✉

*Cryptosporidium* infects gastrointestinal epithelium and is a leading cause of infectious diarrhea and diarrheal-related death in children worldwide. There are no vaccines and no fully effective therapy available for the infection. Type II and III interferon (IFN) responses are important determinants of susceptibility to infection but the role for type I IFN response remains obscure. *Cryptosporidium parvum* virus 1 (CSpV1) is a double-stranded RNA (dsRNA) virus harbored by *Cryptosporidium* spp. Here we show that intestinal epithelial conditional *Ifnar1$^{-/-}$* mice (deficient in type I IFN receptor) are resistant to *C. parvum* infection. CSpV1-dsRNAs are delivered into host cells and trigger type I IFN response in infected cells. Whereas *C. parvum* infection attenuates epithelial response to IFN-γ, loss of type I IFN signaling or inhibition of CSpV1-dsRNA delivery can restore IFN-γ-mediated protective response. Our findings demonstrate that type I IFN signaling in intestinal epithelial cells is detrimental to intestinal anti-*C. parvum* defense and *Cryptosporidium* uses CSpV1 to activate type I IFN signaling to evade epithelial antiparasitic response.

*Cryptosporidium*, an obligate intracellular apicomplexan parasite that infects mammalian gastrointestinal epithelium, is one of the leading pathogens responsible for moderate-to-severe diarrhea in children, particularly under the age of two, and an important contributor to early childhood mortality[1,2]. *Cryptosporidium* is also an important opportunistic pathogen of immunocompromised patients and those undergoing immunosuppressive therapy[3]. There are no vaccines and no fully effective therapy available for the infection. Current understanding of *Cryptosporidium* biology and molecular mechanisms of parasite-host interactions is still obscure, hampering the development of effective therapeutic strategies[4–6]

Humans are infected by ingesting *Cryptosporidium* oocysts, mainly the *C. parvum* and *C. hominis* species[3]. After excystation in the gastrointestinal tract to release infective sporozoites, each sporozoite then attaches to the membrane at their tip of intestinal epithelial cells (mainly enterocytes) and forms an intracellular but extra-cytoplasmic parasitophorous vacuole at the brush border[3,5]. Within the intestine, the infection is limited to epithelial cells and thus, intestinal epithelial cells provide the first line of defense and play a critical role in the initiation, regulation, and resolution of both innate and adaptive immune reactions[6].

The interferon (IFN) family can generally be classified into three main types: type I (e.g., IFN-α and IFN-β), type II (IFN-γ), and type III

[1]Department of Microbial Pathogens and Immunity, Rush University Medical Center, Chicago, IL, USA. [2]Department of Medical Microbiology and Immunology, Creighton University School of Medicine, Omaha, NE, USA. [3]Pediatric Gastroenterology, Children's Hospital & Medical Center, University of Nebraska Medical Center, Omaha, NE, USA. [4]Department of Anatomy and Cell Biology, Rush University Medical Center, Chicago, IL, USA. [5]Animal Parasitic Diseases Laboratory, Agricultural Research Service, the United States Department of Agriculture, Beltsville, MD, USA. [6]Department of Chemistry and Biochemistry, Creighton University College of Arts and Sciences, Omaha, NE, USA. ✉e-mail: xian_m_chen@rush.edu

(IFN-λ family)[7]. Type I IFNs bind to the conserved receptor IFNAR1/2 to induce transcription of type I IFN-stimulated genes[8]. Although type III IFNs bind to the IFNLR1/IL-10RB receptor, they induce highly similar transcriptional responses as type I IFNs[7]. While originally considered to be redundant to type I IFNs, type III IFNs have now been shown to play a unique role in protecting mucosal surfaces against pathogen challenges[9]. Intestinal epithelial cells can produce various type I and III IFNs and act on themselves in an autocrine manner[9]. IFN-γ binds to the heterodimeric receptor of IFNGR1 and IFNGR2 to induce transcription of IFN-γ-stimulated genes[7].

Intestinal epithelial cells themselves do not produce IFN-γ but it is well established that IFN-γ produced from immune cells residing within the epithelium is essential to intestinal innate defense to control *Cryptosporidium* infection[10,11]. Recent studies demonstrated that *Cryptosporidium* can induce a profound type III IFN response in intestinal epithelial cells which is required for early intestinal anti-cryptosporidial defense[12,13]. *Cryptosporidium* infection also induces type I IFN response in the intestinal epithelium[14–16] but there remains controversy about the role for type I IFNs in intestinal anti-*Cryptosporidium* defense. Induced levels of various type I IFN, such as IFN-β1 and IFN-α, are usually very mild in cultured intestinal epithelial cells following *Cryptosporidium* infection[13]. Although pre-treatment of epithelial cells with IFN-α before exposure to *C. parvum* could modestly inhibit parasite invasion of host cells in vitro[14], type I IFN signaling does not protect *C. parvum* infection in IFN-γ knockout (KO) immunocompromised mice[17]. A recent study by Gibson et al. demonstrated that BL6 mice lacking the type I IFN receptor, *Ifnar1*[−/−], consistently resulted in a significantly reduced parasite shedding following *C. parvum* oocysts oral administration when compared to wild-type mice[12], providing strong evidence that type I IFN signaling functions differently from type II/III IFN signaling in host anti-*Cryptosporidium* defense. In addition, to counteract IFN-γ-mediated defense responses, *C. parvum* has evolved strategies to dysregulate IFN-γ signaling in infected host cells[5,18,19] and the underlying mechanism remains obscure.

*Cryptosporidium parvum virus 1* (CSpV1), a member of the family *Partitiviridae*, genus *Cryspovirus*, is a double-stranded RNA (dsRNA) virus of *C. parvum* and many other *Cryptosporidium* spp[20]. CSpV1 virus is non-enveloped with a genome comprised of two distinct dsRNAs, designated as CSpV1-dsRdRp (1836 bp) and CSpV1-dsCA (1510 bp)[21]. CSpV1-dsRNAs are not 5′-capped and unlikely to be 3′-polyadenylylated[20,22]. Herein, we report that conditional intestinal epithelial *Ifnar1* KO neonatal mice are resistant to *C. parvum* infection. Moreover, *C. parvum* infection triggers type I IFN response in intestinal epithelium and attenuates IFN-γ-mediated intestinal epithelial anti-parasitic defense through delivery of CSpV1-dsRNAs. Our data indicate a detrimental effect of type I IFN signaling in intestinal epithelial cells on intestinal anti-*C. parvum* defense and reveal an important strategy of immune evasion by the parasite.

## Results

### Resistance to *C. parvum* infection in vivo in conditional intestinal epithelial *Ifnar1* KO mice

We cross-bred the *Ifnar1*[fl/fl] mice [B6(Cg)-*Ifnar1*[tm1.1Ees]/J, Jackson Lab] with the *Villin.Cre* transgenic [B6.Cg-Tg(Vil1-cre)1000Gum/J, Jackson Lab] and generated conditional intestinal epithelial *Ifnar1* KO mice (*Villin.Ifnar1*[−/−] mice) (Supplementary Fig. 1a, b). *Villin.Ifnar1*[−/−] mice grow normally and the *Villin.Ifnar1*[−/−] neonates showed similar ileal morphological features as the *Ifnar1*[fl/fl] littermates, including HE staining (Fig. 1a), epithelial expression of the epithelial cellular adhesion molecule (EpCAM) and villin, and the presence of proliferating cells (PCNA-positive) and lysozyme-positive Paneth cells in the crypts by immunostaining (Supplementary Fig. 1b, c).

Neonates (5-day old, male and female animals were randomly used) of the cross-bred mice were infected by oral administration of *C. parvum* oocysts[23,24]. Ileal tissues (4 cm of small intestine from the ileocecal junction) were collected at 48 h and 72 h post infection (p.i.) either for morphologic observation or for isolation of ileal epithelium. Parasite burden was measured using real-time quantitative PCR (RT-qPCR) with primers specific for *cp18S* (Fig. 1b) and *cpHsp70* (Supplementary Fig. 1d) and by counting after immunofluorescent staining (Fig. 1b, c)[25,26]. A significantly lower infection burden was detected in the ilea from the *Villin.Ifnar1*[−/−] mice than the *Ifnar1*[fl/fl] control (Fig. 1a-c and Supplementary Fig. 1b–d). A decrease in ileal villus heights, accompanied with increased cell proliferation (PCNA-positive) and expansion of the crypt regions, was observed in both infected *Villin.Ifnar1*[−/−] and *Ifnar1*[fl/fl] mice compared with non-infected animals (Fig. 1c). Epithelial labeling of EpCAM (Fig. 1c) and villin, and crypt distribution of Paneth cells (Supplementary Fig. 1c) were similar in infected animals of both genotypes.

We then measured *C. parvum* infection burden in cultured IEC4.1 cells (transformed but non-tumorigenic intestinal epithelial cells from neonatal mice)[27] and stable IEC4.1 cells deficient in *Ifnar1* (IEC4.1-*Ifnar1*[−/−] cells) (Supplementary Fig. 2). Parasite burden at 24 h after exposure to *C. parvum* (a time point reflecting activating of the cellular defenses protecting against infection)[16], as well as at 2 h after exposure to parasite (a time point reflecting the initial attachment and invasion of the parasite into host cells)[16,28], was similar between IEC4.1 and IEC4.1-*Ifnar1*[−/−] cells (Fig. 1d). Thus, conditional intestinal epithelial *Ifnar1*[−/−] mice develop resistance to *C. parvum* intestinal infection (Fig. 1b). However, in vitro infection employing *Ifnar1* KO intestinal epithelial cells showed no changes in *C. parvum* attachment/invasion or epithelial cell antiparasitic defense.

### Alterations in gene expression profiles, including various types of IFNs and IFN-stimulated genes, in the intestinal epithelium or cell cultures following *C. parvum* infection

We first measured the expression levels of selected type I, II, and III IFN genes in the neonatal intestinal epithelium and in IEC4.1 cells following infection. The RNA levels of several type I IFNs (*Ifna13*, *Ifna7*, and *Ifnb1*), as well as type II *Ifng* and type III *Ifnl2/3*, were significantly increased in the intestinal epithelium at 48 h p.i. (Fig. 2a). The RNA levels of many type I IFN genes, including *Ifna13*, *Ifna7*, *Ifna1/5/6*, *Ifna1/2/5*, *Ifna4*, *Ifnab*, *Ifna12*, and *Ifnb1*, were significantly increased in IEC4.1 cells 24 h post-infection (Fig. 2b). However, *Ifng* RNA was not detected in either infected or non-infected IEC4.1 cells and no significant increase of *Ifnl2/3* at the RNA level was detected in IEC4.1 cells at 24 h after infection (Fig. 2b). The protein level of IFN-β1 was measured and confirmed to be increased in the supernatants of infected IEC4.1 cultures (24 h and 48 h p.i.) (Fig. 2b). The protein levels of IFN-γ and IFN-λ2/3 in non-infected and infected IEC4.1 cells were both below the detectable limit of the ELISA kit (DY485 and DY1789D from the R & D Systems).

We performed the RNA-Seq analysis of the ileal epithelium from infected *Villin.Ifnar1*[−/−] and *Ifnar1*[fl/fl] mice (48 h and 72 h p.i.). The complete list of genes differentially expressed between uninfected and infected *Ifnar1*[fl/fl] or *Villin.Ifnar1*[−/−] mice were listed in Supplementary Data 1. For the *Ifnar1*[fl/fl] mice, a total of 2120 and 3,665 protein-coding genes showed a significantly different expression level (adjusted $p < 0.05$ with fold change either >1.5 upregulated or <0.5 down-regulated) between uninfected and infected neonates at 48 h and 72 h p.i., respectively (Supplementary Data 1). For the *Villin.Ifnar1*[−/−] mice, expression levels of 592 and 1142 genes were significantly different ($p < 0.05$) between uninfected and infected animals (at 48 h and 72 h p.i., respectively) (Supplementary Data 1). Comparison of differentially expressed genes between the two infected genotypes revealed that a majority of genes whose expression levels were altered in infected *Ifnar1*[fl/fl] mice (78.25% and 71.54% of total genes with an altered expression level at 48 h and 72 h p.i., respectively) showed no changes in *Villin.Ifnar1*[−/−] neonates at the same timepoint after infection (Fig. 2c), suggesting a strong type I IFN-associated gene transcription in the intestinal epithelium of the wild-type *Ifnar1*[fl/fl] neonates following

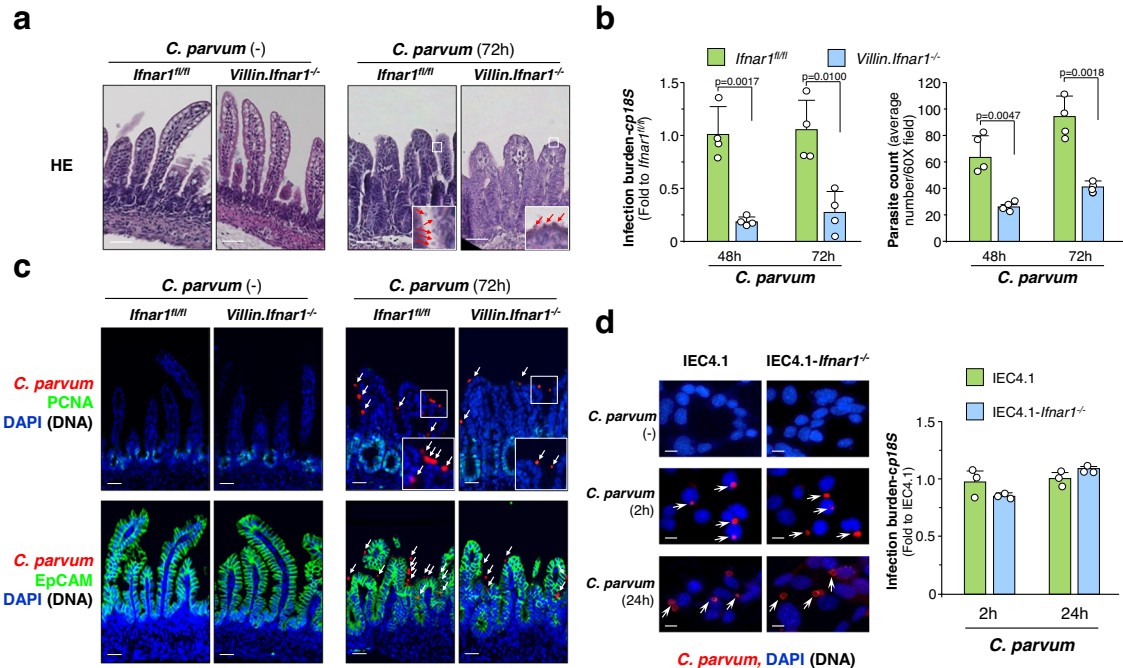

**Fig. 1 | *C. parvum* infection in conditional intestinal epithelial *Ifnar1* knockout mice and IEC4.1 cell cultures. a** HE morphological features of small intestine of *Ifnar1^fl/fl* and *Villin.Ifnar1^−/−* neonates following infection. Neonatal mice (5 days old) were orally inoculated with *C. parvum* oocysts (10^6 oocysts per animal) and ileal epithelium (4 cm of small intestine tissue from the ileocecal junction) was collected (72 h p.i.) Representative HE images from four independent experiments are shown. Insets are higher magnification of the boxed regions showing the parasites (arrows). Bars = 50 μm. **b** *C. parvum* intestinal infection burden in *Ifnar1^fl/fl* and *Villin.Ifnar1^−/−* mice. Neonatal mice were inoculated with *C. parvum* oocysts (10^6 oocysts per animal) for 48 h and 72 h. Infection burden was evaluated by RT-qPCR of *cp18S* gene in the isolated intestinal epithelium (fold changes to *Ifnar1^fl/fl* normalized to host *Gapdh*) and parasite counting of intestine sections after immunofluorescent staining of *C. parvum* (average number/60X field). Data are presented as mean values ± SD, compiled from 4 independent experiments and the

dots represent the mean value of each experiment with 6 mice in each group. Statistical significance was determined by two-tailed unpaired Student's *t*-test. **c** Immunofluorescent staining of *C. parvum*, and PCNA and EpCAM staining of small intestine of *Ifnar1^fl/fl* and *Villin.Ifnar1^−/−* neonates with and without *C. parvum* infection (at 72 h p.i.). Representative images from 4 independent experiments are shown. Insets are higher magnification of the boxed regions showing the parasites revealed by immunofluorescent staining for parasite counting (arrowheads). Blue: DAPI (DNA), green: PCNA or EpCAM, red: *C. parvum* (arrows). Bars = 50 μm. **d** No significant difference in infection burden between IEC4.1 and IEC4.1-*Ifnar1^−/−* cells following *C. parvum* infection in vitro. IEC4.1 and IEC4.1-*Ifnar1^−/−* cells were infected with *C. parvum* for 2 h and 24 h. Infection burden was evaluated by RT-qPCR (*cp18S*, fold changes to IEC4.1 normalized to *Gapdh*). The dots represent data from three biological replicates. Data are presented as mean values ± SD. Blue: DAPI (DNA), red: *C. parvum* (arrows). Bars = 5 μm. Source data are provided as a Source Data file.

infection. Gene set enrichment analysis (GSEA) of the gene expression profile in the infected *Ifnar1^fl/fl* mice revealed various biological pathways associated with infection, including upregulation of genes for the type I IFN production and cellular response to type I IFNs, IFN-γ-mediated signaling and cytokine-mediated signaling pathways (Fig. 2d), consistent with findings from previous studies[13,15,29]. Induction of the type I/III IFN-stimulated genes was generally suppressed in the ilea from infected *Villin.Ifnar1^−/−* mice (Supplementary Data 1), particularly at 48 h p.i. (Fig. 2d–f). Specifically, GSEA revealed a significant decrease in cellular response to type I IFNs in the infected *Villin.Ifnar1^−/−* mice compared with the infected wild-type *Ifnar1^fl/fl* neonates. In contrast, the expression levels of many IFN-γ-stimulated genes were comparable between infected *Villin.Ifnar1^−/−* and *Ifnar1^fl/fl* animals (Fig. 2d–f). Interestingly, upregulation of 18 genes, including *Adora2b, Mttp, Arhgap26, Btnl6, Erbb2, Lipe,* and *Vmp1*, was further significantly enhanced in the infected ilea of *Villin.Ifnar1^−/−* mice (48 h p.i.), compared with infected *Ifnar1^fl/fl* littermates (Fig. 2e, f). This panel of genes were represented as the "miscellaneous (misc.) upregulated genes". Nevertheless, the expression levels of most other inflammatory genes or genes that code proteins for cell proliferation and apoptosis were comparable between infected *Villin.Ifnar1^−/−* and *Ifnar1^fl/fl* animals (Fig. 2d and Supplementary Data 1), which may account for the similar morphological features and intestinal damage observed between infected *Villin.Ifnar1^−/−* and *Ifnar1^fl/fl* groups (Fig. 1a, c).

RNA-Seq analysis revealed significant alterations in gene expression levels in IEC4.1 and IEC4.1-*Ifnar1^−/−* cells at 24 h p.i. (Supplementary

Fig. 3a and Supplementary Data 2). GSEA revealed several biological pathways associated with infection in infected IEC4.1 cells, including increase of type I IFN production and cellular response to type I IFNs and cytokine-mediated signaling pathway (Supplementary Fig. 3b). However, no significant increase of signaling associated with cellular response to IFN-γ was observed in infected IEC4.1 cells (Supplementary Fig. 3b). Comparison of differentially expressed genes between infected IEC4.1 and IEC4.1-*Ifnar1^−/−* cells show that a majority of genes whose expression levels were altered in infected IEC4.1 cells (73.01% of total genes with an altered expression level at 24 h p.i.) showed no changes in the infected IEC4.1-*Ifnar1^−/−* cells (Fig. 2g), suggesting a strong type I IFN-associated gene transcription in IEC4.1 cells following infection in vitro. Indeed, GSEA revealed that signaling pathways associated with cellular response to type I IFNs was blocked in infected IEC4.1-*Ifnar1^−/−* cells (Fig. 2h). Upregulation of type I/III IFN-stimulated genes induced by infection was generally suppressed in the infected *IEC4.1-Ifnar1^−/−* cells (Supplementary Fig. 3c). In addition, no significant induction of these "misc. upregulated genes" was observed in IEC4.1 and IEC4.1-*Ifnar1^−/−* cells following infection (Supplementary Fig. 3c). Expression levels of selected genes associated with various types of IFNs in in vitro infected cell cultures or intestinal epithelium from in vivo *C. parvum* infection were further validated at the RNA level using RT-qPCR (for *Ifnb1, Ifi44, Iigp1, Ifng, Ido1, Ifn1/2/5, Oas1g,* and *Usp18;* Supplementary Fig. 3d, e) and at the protein level by Western blot (for Isg15 and Usp18, Supplementary Fig. 3f). The above data demonstrate differential gene expression profiles between infected intestinal epithelium from

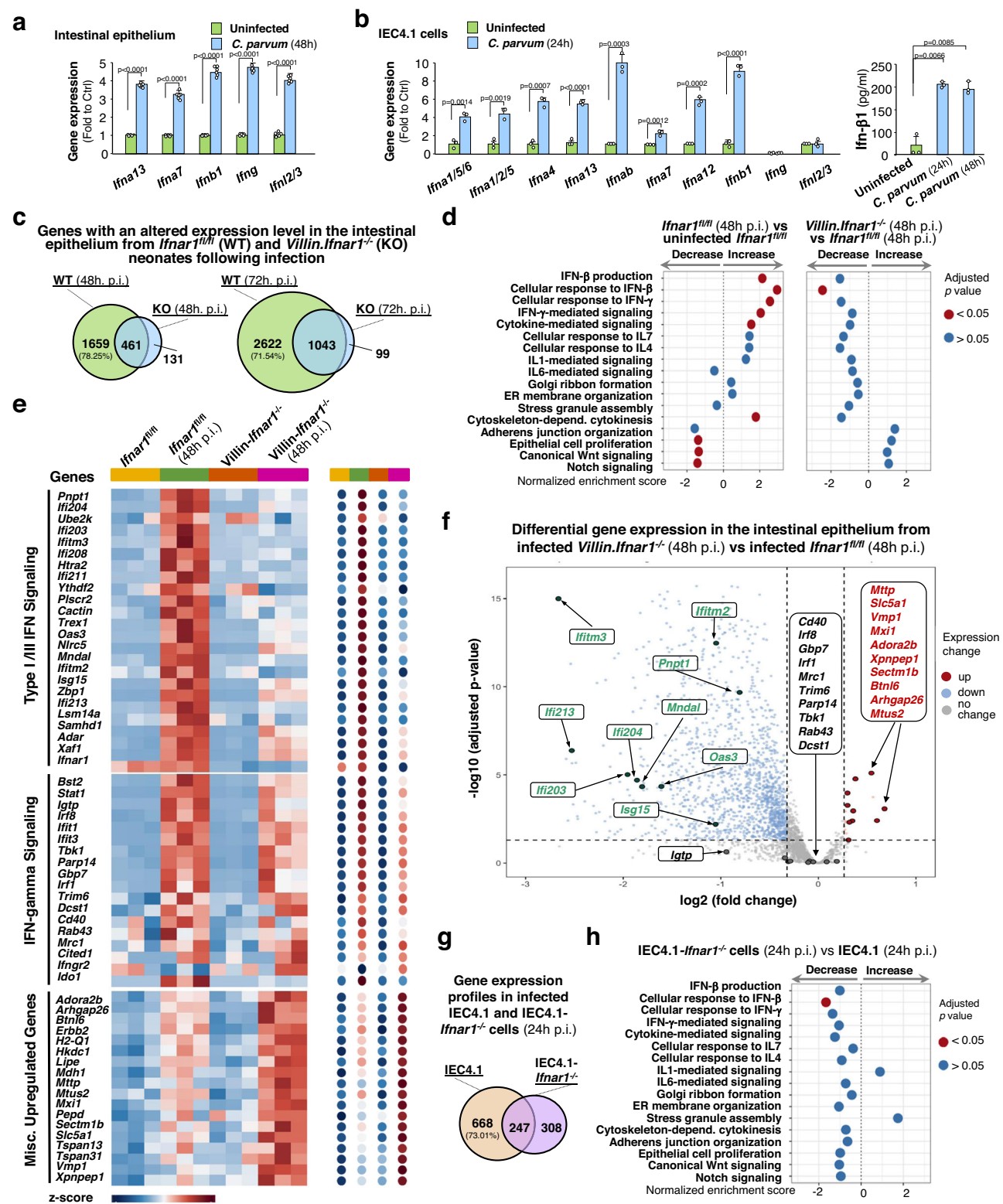

*Villin.Ifnar1⁻/⁻* and *Ifnar1^{fl/fl}* neonates or infected IEC4.1 and IEC4.1-*Ifnar1⁻/⁻* cells, suggesting a strong type I IFN response in intestinal epithelial cells following *Cryptosporidium* infection.

**Delivery of CSpV1-dsRNAs into infected host cells following *C. parvum* infection**

Molecules of parasite origin can selectively be delivered into host cells following *C. parvum* infection[30–32]. The genome of CSpV1 comprises

two distinct dsRNAs, CSpV1-dsRdRp and CSpV1-dsCA (Supplementary Fig. 4a). Using RT-qPCR, we detected a high level of CSpV1-dsRdRp and CSpV1-dsCA in the cytoplasmic fraction of infected IEC4.1 cells (Fig. 3a). In contrast, several parasite RNA transcripts abundantly expressed in *C. parvum* during the early infection stage[33,34] were not detected in the cytoplasmic fraction from infected cells (Fig. 3a). Using in situ hybridization, we confirmed the cytoplasmic presence of CSpV1-dsRdRp in infected IEC4.1 cells (Fig. 3b). Notably, neither

**Fig. 2 | Gene expression profiles in infected intestinal epithelium and cell cultures. a** Expression of IFNs in intestinal epithelium from infected *Ifnar1*[fl/fl] neonates. Mice (5 days old) were orally inoculated with *C. parvum* ($10^6$ oocysts/animal) and ileal epithelium (4 cm from the ileocecal junction) was isolated at 48 h p.i. IFN gene expression was evaluated by RT-qPCR and presented as fold change to uninfected control. Dots represent data from experiment with six mice in each group and presented as mean values ± SD. **b** Expression of IFNs in infected IEC4.1 cells. IFN gene expression (24 h p.i.) was evaluated by RT-qPCR. Ifn-β1 protein content in the supernatants (24 h and 48 h p.i.) was measured by ELISA. Data are from three biological replicates and presented as mean values ± SD. *p* values were determined by two-tailed unpaired Student's *t*-test (in **a** and **b**). **c** Gene expression profiles in intestinal epithelium from infected neonates (48 h and 72 h p.i.) by RNA-Seq. The numbers of genes with an altered expression level following infection are listed. **d** Gene set enrichment analysis (GSEA) in intestinal epithelium from infected *Ifnar1*[fl/fl] (48 h p.i.) compared with that in uninfected *Ifnar1*[fl/fl] or infected *Villin.Ifnar1*[−/−] neonates (48 h p.i.). *p* values were calculated based on Kolmogorov−Smirnov test and adjusted by Benjamini−Hochberg method. Normalized enrichment scores and adjusted *p* values for the signaling pathways are

shown. **e** Selected gene groups revealed by RNA-Seq in intestinal epithelium from infected *Ifnar1*[fl/fl] and *Villin.Ifnar1*[−/−] neonates (48 h. p.i.). Representative genes for type I/II/III IFN signaling and misc. upregulated genes (expression levels further increased in infected *Villin.Ifnar1*[−/−] neonates) are shown. **f** Volcano plots depicting the differentially upregulated genes between infected *Villin.Ifnar1*[−/−] and *Ifnar1*[fl/fl] animals (48 h p.i.). Two-tailed Wald tests were performed for statistical analysis and *p* value was adjusted by Benjamini−Hochberg method. Dashed line indicates a false discovery rate cutoff of 0.05. Data are from three animals for each group (**c**−**f**). **g** Gene expression profiles in IEC4.1 following infection. Numbers of genes whose expression levels were significantly altered in infected IEC4.1 and IEC4.1-*Ifnar1*[−/−] cells (24 h p.i.) by RNA-Seq are listed. **h** GSEA of gene expression profiles in infected IEC4.1-*Ifnar1*[−/−] cells (24 h p.i.) compared with that in infected IEC4.1 cells (24 h p.i.). Data are from three biological replicates (3 RNA-seq replicates each group) in **g** and **h**. *p* values (in **h**) were calculated based on Kolmogorov−Smirnov test and adjusted by Benjamini−Hochberg method. Normalized enrichment scores and adjusted *p* values for the signaling pathways are shown. Source data are provided as a Source Data file.

CSpV1-dsRdRp nor CSpV1-dsCA was detected in cells after exposure to heat-killed parasites (Supplementary Fig. 4b), indicating that CSpV1-dsRNA delivery is not due to nonspecific endocytosis.

J2 is a monoclonal antibody developed to recognize global viral dsRNAs[35]. Using the J2 antibody for immunofluorescent staining, we observed positive signal in the cytoplasm of infected IEC4.1 cells or cells after transfection of CSpV1-dsRNAs (Fig. 3c). Such positive labeling was not observed in the uninfected cells or cells transfected with a non-specific scrambled double-stranded siRNA (Control RNA) (Fig. 3c). Dot blotting also detected in vitro-transcribed CSpV1-dsRNAs (Fig. 3d) and revealed the presence of viral dsRNAs in the lysate of *C. parvum* sporozoites (Fig. 3e) and the cytoplasmic fraction of infected IEC4.1 cultures or cells transfected with CSpV1-dsRNAs (Fig. 3f). An RNA immunoprecipitation (RIP) assay was performed using the J2 antibody and the presence of CSpV1-dsRdRp and CSpV1-dsCA was detected in the immunoprecipitates from infected cells or cells transfected with in vitro transcribed CSpV1-dsRdRp or CSpV1-dsCA (Fig. 3g). Selected RNAs of IEC4.1 cells origin (Gapdh and Actin) and parasite origin (Cgd7_FLc_0990 and Cgd7_FLc-1000)[32] were not detected in the immunoprecipitates using J2 (Supplementary Fig. 4c), confirming the specificity of J2 antibody.

**Host delivery of CSpV1-dsRNAs triggers activation of type I IFN signaling in infected cells**
We first examined type I IFN response in IEC4.1 cells following transfection with in vitro transcribed CSpV1-dsCA or CSpV1-dsRdRp alone. Upregulation of type I IFN and type I IFN-stimulated genes was observed in cells transfected with either CSpV1-dsRdRp or CSpV1-dsCA alone (Fig. 4a). Transfection of the same amount of single-stranded RNA of CSpV1-RdRp or CSpV1-CA (either the forward or reverse) caused only a minor induction of type I IFN-stimulated genes (Fig. 4b), less than one-tenth of the levels compared with CSpV1-dsRdRp or CSpV1-dsCA.

We then performed RNA-Seq analysis in IEC4.1 cells after transfection of CSpV1-dsRdRp, CSpV1-dsCA, or in combination for 24 h and compared their gene expression profiles with that in infected IEC4.1 cells (24 h p.i.). Significant alterations in the gene expression profiles were observed in cells transfected with CSpV1-dsRdRp, CSpV1-dsCA, or together, compared with cells transfected with a non-specific scrambled double-stranded siRNA (Control RNA) (Fig. 4c, d). A comparison of genome-wide transcriptomics indicated that genes whose expression levels altered in cells transfected with either CSpV1-dsRdRp, or CSpV1-dsCA, or together overlaid partially with that in cells following *C. parvum* infection (Fig. 4c, d). Specifically, similar to cells following infection (24 h p.i.), expression levels of many type I IFN-stimulated genes were increased in cells after transfection of either CSpV1-dsRdRp or CSpV1-dsCA alone or in combination for 24 h

(Fig. 4e). A complete list of genes differentially expressed in cells transfected with Control RNA, CSpV1-dsRdRp, CSpV1-dsCA, or together was listed in Supplementary Data 3. Moreover, we developed an approach to interfere with CSpV1-RNA delivery through transfection of host cells with the siRNA combinations (siPOOLs) designed to target CSpV1-dsRNAs, because transfection of *C. parvum* parasite with any plasmid remains a technical challenge if not impossible[3,36]. Compared with infected cells treated with the siRNA Control (siControl), IEC4.1 cells transfected with siPOOLs to CSpV1-dsRNAs following infection showed a decreased level of CSpV1-RdRp level (Fig. 4f) with a similar infection burden (Fig. 4g) but a significant inhibition on *C. parvum*-induced *Ifnb1* expression (Fig. 4h).

**CSpV1-dsRNAs induce type I IFN response in infected cells through activating the Pkr and Rig-I/Mavs/Sting signal pathways**
Viral dsRNAs are recognized by intracellular dsRNA sensors to trigger type I IFN gene transcription, such as the interferon-inducible protein kinase R (PKR), retinoic acid-inducible gene I (RIG-I), and the melanoma differentiation-associated gene 5 (MDA5)−mitochondrial antiviral signaling protein (MAVS)−stimulator of IFN genes (STING) pathways, and Toll-like receptor 3 (TLR3)[37]. We used the CRISPR/Cas9 approach to generate stable IEC4.1 cells deficient in *Ddx58* (encoding Rig-I), *Ifih1* (encoding Mda5), *Mavs*, or *Eif2ak2* (encoding Pkr) (Fig. 5a). We took the siRNA approach to knockdown *Tlr3* in IEC4.1 cells (Fig. 5b). Knockout of *Ddx58*, *Mavs*, or *Eif2ak2* in IEC4.1 cells completely blocked *C. parvum*- and CSpV1-dsRNA-induced upregulation of *Ifnb1* (Fig. 5c). Knockout of *Ifih1* or siRNA knockdown of *Tlr3* failed to block *C. parvum*- and CSpV1-dsRNA-induced *Ifnb1* expression (Fig. 5c). Moreover, inhibition of Sting signaling by the inhibitor C-178[38] or Pkr signaling by C-16[39] partially blocked *C. parvum*- and CSpV1-dsRNA-induced *Ifnb1* expression (Fig. 5d). Using the crosslinking RNA-binding protein immunoprecipitation (RIP) assay, we detected a physical association between CSpV1-dsRNAs with Pkr, Rig-I, or Mavs in cells transfected with CSpV1-dsRNAs (Fig. 5e) or cells following *C. parvum* infection for 24 h (Fig. 5f). Of note, although knockout of *Ifih1* failed to block *C. parvum*- and CSpV1-dsRNA-induced *Ifnb1* expression (Fig. 5c), we still detected a physical association between CSpV1-dsRNAs with Mda5 in infected cells or cells transfected with CSpV1-dsRNAs (Fig. 5e, f). These findings suggest that *C. parvum* infection triggers host type I IFN response through CSpV1-dsRNA-mediated activation of Pkr and Rig-I/Mavs/Sting signaling (Fig. 5g).

**Suppression of cellular response to IFN-γ stimulation with the involvement of type I IFN signaling in infected host cells**
Pretreatment of IEC4.1 cells with IFN-γ promoted host defense to *C. parvum* infection (Fig. 6a). Consistent with previous reports[24,40], we

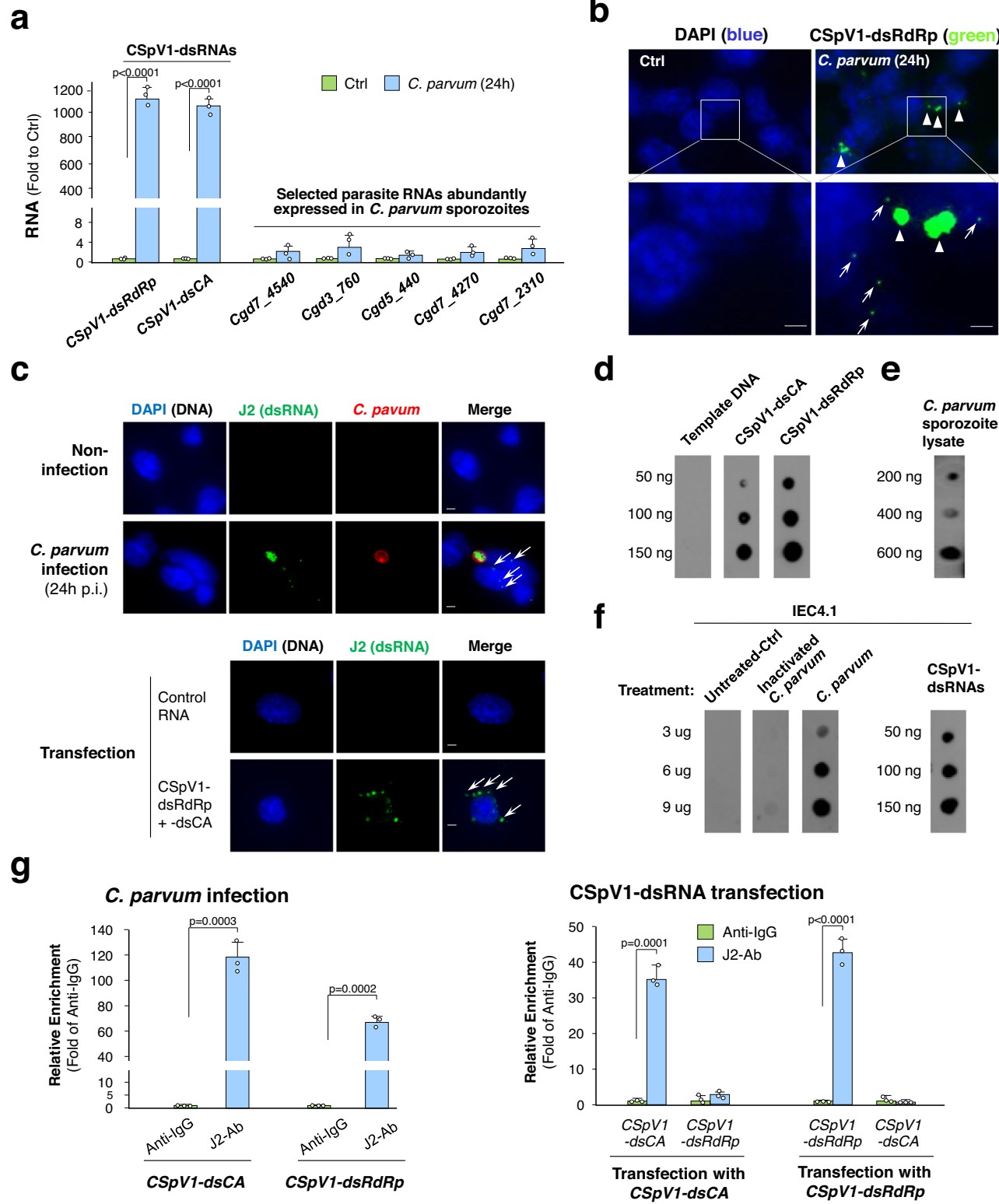

detected an impaired cellular response to IFN-γ in cells after *C. parvum* infection. Compared with that in uninfected cells, a decrease in IFN-γ-induced expression of *Ido1* was observed in IEC4.1 cells first exposed to *C. parvum* for 24 h followed by IFN-γ stimulation for additional 2 h (Fig. 6b). Such a decreased IFN-γ response was not observed in infected *Ifnar1⁻/⁻* IEC4.1 cells (Fig. 6b). Knockdown of CSpV1-dsRNA delivery in infected IEC4.1 cells, through host cell transfection of the specific siRNA combinations (siPOOLs), restored the induction of *Ido1* gene expression in host cells following *C. parvum* infection (Fig. 6c).

RNA-Seq analysis revealed no significant changes in the expression levels of key components associated with the IFN-γ signal pathway, including downstream JAK/STAT signaling between infected IEC4.1 and *Ifnar1⁻/⁻* IEC4.1 cells, as well as ileal epithelium from infected *Ifnar1fl/fl* and *Villin.Ifnar1⁻/⁻* neonates (Fig. 6d). A similar protein content of the IFN-γ receptor Ifngr1 subunit was detected by Western in IEC4.1 and IEC4.1-*Ifnar1⁻/⁻* cells (Fig. 6e); ileal epithelium of *Ifnar1fl/fl* and *Villin.Ifnar1⁻/⁻* neonates showed a similar protein content (Fig. 6e) and a similar staining pattern for Ifngr1 (Fig. 6f). Decreased expression levels

**Fig. 3 | Delivery of CSpV1-dsRNAs into the cytoplasmic region of intestinal epithelial cells following *C. parvum* infection. a** Detection of CSpV1-dsRNAs in the cytoplasm of infected cells. IEC4.1 cells were exposed to *C. parvum* for 24 h and cytoplasmic fractions were isolated. *CSpV1-dsRdRp, CSpV1-dsCA*, and several parasite RNAs (e.g., *Cgd7_4540*) in the cytoplasmic fractions were measured by RT-qPCR. Data are presented as the fold change to control normalized to host *Gapdh*. Cytoplasmic extract from cells collected from cell suspension mixed with the same amount of parasites was used as control (Ctrl). Data are from three biological replicates and presented as mean values ± SD. *p* values were determined by two-tailed unpaired Student's *t*-test. **b** In situ hybridization of CSpV1-dsRdRp in *C. parvum*-infected cell. IEC4.1 cells were exposed to *C. parvum* for 24 h and CSpV1-dsRdRp was detected in the cytoplasm (arrows) using DIG-labeled CSpV1-dsRdRp probes. Representative images from three independent experiments are shown. Blue: DAPI (DNA), green: CSpV1-dsRdRp (arrows), *C. parvum*: arrowheads. Bars = 1 μm. **c** Immunofluorescent staining of CSpV1-dsRNAs using J-2 antibody in IEC4.1 cells following *C. parvum* infection (24 h) or in cells transfected with CSpV1- dsRdRp, CSPV1-dsCA, or a non-specific double-stranded siRNA (control RNA) for 24 h. Representative images from three independent experiments are shown. Blue: DAPI (DNA), red: *C. parvum*, green: CSpV1-dsRNAs (positive for J2 antibody, arrows). Bars = 1 μm. **d**–**f** Dot blots using J2 antibody to detect CSpV1-dsRNAs. Detection of in vitro-transcribed CSpV1-dsCA and CSpV1-dsRdRp (template plasmid DNA for in vitro transcription as control) (**d**); detection of CSpV1-dsRNAs in different doses of *C. parvum* sporozoite lysate (**e**); detection of CSpV1-dsRNAs in untreated IEC4.1 cells and cells infected by inactivated *C. parvum* or infective *C. parvum* or transfected with CSpV1-dsRNAs (**f**). Representative dot blots from three independent experiments are shown. **g** RIP assay for CSpV1-dsRNAs from IEC4.1 cells following *C. parvum* infection or CSpV1-dsCA or CSpV1-dsRdRp transfection. Cells were exposed to *C. parvum* or transfected with CSpV1-dsCA or CSpV1-dsRdRp for 24 h. CSpV1-dsRNAs were pulled down using J2 antibody. Isotype IgG was used as negative control. Data are from three biological replicates and presented as mean values ± SD. *p* values were determined by two-tailed unpaired Student's *t*-test. Source data are provided as a Source Data file.

of several other selected IFN-γ-stimulated genes, including *Igtp, Irf8*, and *Ifi47*, were further observed in IEC4.1 cells first exposed to *C. parvum* for 24 h followed by IFN-γ stimulation for additional 2 h (Supplementary Fig. 5a). Suppression of expression of *Igtp* and *Irf8* (but not *Ifi47*) was observed in IEC4.1 cells first treated with IFN-α for 8 h followed by IFN-γ stimulation for additional 2 h in the absence of IFN-α (Supplementary Fig. 5b). Together, these data suggest that suppression of cellular response to IFN-γ stimulation with the involvement of type I IFN signaling in infected cells but such a suppressive effect appears not due to the direct effect of type I IFN signaling on the expression of key components of the IFN-γ signaling cascade in host cells.

### Effects of anti-viral reagents and anti-parasite drugs on *C. parvum* infection of intestinal epithelium

Consistent with the results from previous studies[41], a small but significant decrease in the infection burden was detected in cultured human intestinal epithelial HCT-8 cells treated with the antiviral reagents, Indinavir or Ribavirin (Fig. 7a). Cells were first exposed to *C. parvum* infection for 4 h (for parasite attachment and invasion)[16,28] and then exposed to Indinavir or Ribavirin for additional 20 h followed by measurement of infection burden. Antiparasitic reagent paromomycin was used as the positive control[41]. A further decrease in infection burden was observed in cells treated with a combination of paromomycin with Indinavir or Ribavirin (Fig. 7a). Using frozen intestinal tissues from an existing human tissue bank (University of Nebraska Medical Center), we successfully developed 3D enteroids and 2D monolayers for *C. parvum* ex vivo infection (Fig. 7b, c). Induction of type I and III IFNs and Type I/III IFN-stimulated genes was detected in human 2D monolayers following *C. parvum* infection, further confirming the activation of type I and III IFN signaling in infected human intestinal epithelial cells (Fig. 7d). A significant decrease in the infection burden was detected in the human 2D monolayers treated with the antiviral reagents, Indinavir or Ribavirin (Fig. 7e). A further decrease in infection burden was observed in the monolayers treated with a combination of paromomycin with Indinavir or Ribavirin (Fig. 7e). Antiparasitic and antiviral drugs were administered to C57BL/6 neonates of 5 days old at 2 h after oral gavage of *C. parvum* oocysts, followed by a daily administration of the drugs for 3 days. Decreased infection burden was detected in the animals received paromomycin or Ribavirin, but not Indinavir (Fig. 7f, g). A further decrease in infection burden was observed in the intestinal epithelial tissues from animals received a combination of paromomycin and Ribavirin (Fig. 7f, g).

## Discussion

Our data demonstrate that *Cryptosporidium* infection induces all type IFN responses in the intestinal epithelium in neonatal mice, including production of three types of IFNs and upregulation of many type I/III IFN- and IFN-γ-stimulated genes. We also detected increased expression of several type I IFNs in the intestinal epithelial cell cultures infected in vitro. More importantly, a majority of genes whose expression levels were altered in infected *Ifnar1*$^{fl/fl}$ mice showed no changes in the infected conditional intestinal epithelial cell *Villin-Ifnar1*$^{-/-}$ neonates, further supporting a strong type I IFN-mediated gene transcription in the infected intestinal epithelium. Type I and III IFNs bind to different receptors but activate a similar intracellular signaling cascade and induce highly similar transcriptional responses[7,8]. Suppression of type I/III-stimulated gene upregulation in intestinal epithelium isolated from the infected *Villin*-Ifnar1$^{-/-}$ mice also suggests that type I IFN-stimulated gene transcription in intestinal epithelial cells could not fully be complemented by type III IFNs. Thus, *Cryptosporidium* infection can induce type I IFN produce from intestinal epithelial cells and activation of type I IFN signaling in intestinal epithelial cells contributes significantly to the transcriptomic alterations in the infected intestinal epithelium, including transactivation of many type I/III-stimulated genes, at least during the early infection stage in neonatal mice.

Current understanding of the role for type I IFN signaling in intestinal anti-*Cryptosporidium* defense remains obscure. Previous studies demonstrated that pre-treatment of epithelial cells with IFN-α before exposure to *C. parvum* could modestly inhibit parasite invasion of host cells in vitro[14]. Type I IFN response reduced oocyst shedding and symptoms but not intestinal parasite burden following *C. parvum* infection in IFN-γ KO mice[17]. Gibson et al. recently reported that *Cryptosporidium* shedding was significantly reduced in the *Ifnar1*$^{-/-}$ mice compared to wild-type animal[12]. Whether this resistance to infection is due to loss of type I IFN signaling in intestinal epithelial cells or in other cell types such as T cells is unclear. Our findings support that intestinal epithelial conditional *Ifnar1*$^{-/-}$ mice are resistant to *C. parvum* infection, indicating a detrimental effect of type I IFN signaling in intestinal epithelial cells on intestinal anti-*C. parvum* defense. Therefore, although all three types of IFN signaling are activated in the neonatal intestinal epithelium following infection, each type of IFNs displays distinct roles in intestinal immunity to *Cryptosporidium* infection. Whereas type II and III IFNs show a profound protective role in control the infection, type I IFN signaling in the intestinal epithelial cells displays a pro-parasitic effect potentially through suppression of IFN-γ-mediated epithelial cell anti-*Cryptosporidium* defense.

Our observation of resistance to *C. parvum* intestinal infection in conditional intestinal epithelial *Ifnar1*$^{-/-}$ neonatal mice, but not in vitro infection in cultured cells lacking *Ifnar1*, suggests that type I IFN signaling may modulate intestinal anti-*Cryptosporidium* defense induced by determinants from other cell types during infection in vivo. One possible mechanism could be the potential impact of type I IFN response on type III IFN-mediated epithelial defense. Of note, no obvious induction of *Ifnl2/3* genes and produce of IFN-λ2/3 proteins

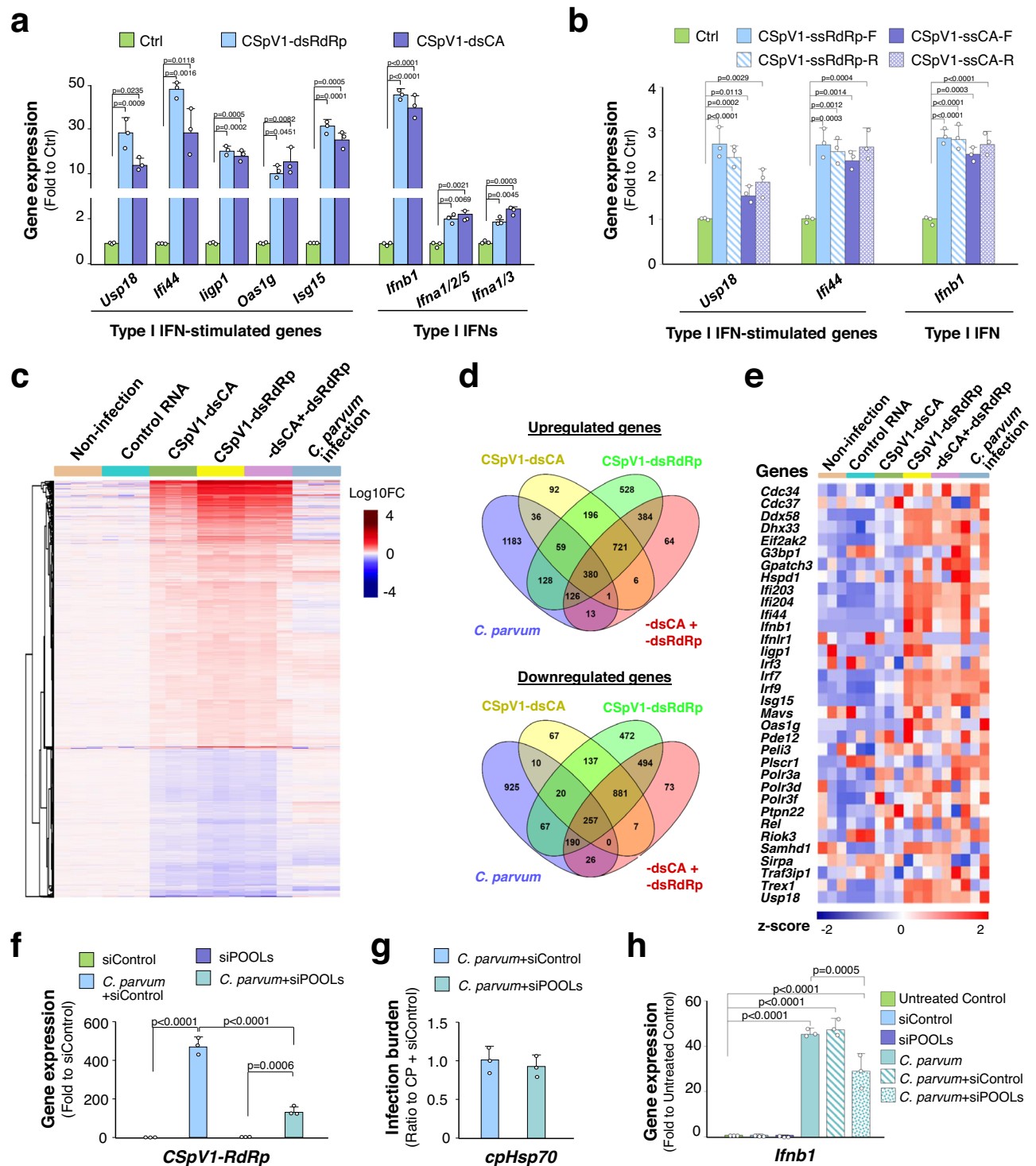

were observed in the IEC4.1 cells cultures following infection in vitro. While originally considered to be redundant to type I IFNs, type III IFNs have now been shown to play a unique role in protecting mucosal surfaces against pathogen challenges[7]. Our findings provide additional evidence that type I and III IFNs have distinct functions in intestinal epithelial cells, even though they stimulate a similar intracellular signaling cascade and an indistinguishable gene expression profile in the cells[9]. Indeed, type III IFNs show a profound protective role in control *Cryptosporidium* infection as mice lacking the type III receptor, *Ifnlr1*, showed an increase of parasite shedding following infection[12]. Future investigations on *Cryptosporidium* infection using conditional intestinal epithelial *Ifnlr* KO animals should clarify these possibilities.

Previous studies have demonstrated a dysregulation of IFN-γ-mediated signaling pathway in intestinal epithelial cells by *Cryptosporidium parvum* infection[5,18,19]. Our data support that activation of type I IFN signaling may be involved in *C. parvum*-induced dysregulation of IFN-γ response in intestinal epithelial cells. Whereas parasite attachment and host cell invasion were similar in cultured IEC4.1 and IEC4.1-*Ifnar1*[−/−] cells, we detected an impaired response to IFN-γ stimulation in infected IEC4.1 cells but not infected IEC4.1-*Ifnar1*[−/−] cells. Similarly, pre-treatment of cells with type I IFNs (IFN-α) also attenuated the cellular response to IFN-γ stimulation. Thus, we favor the hypothesis that activation of type I IFN signaling following *C. parvum* infection may cause dysregulation of IFN-γ-mediated cell response in intestinal

**Fig. 4 | Delivery of CSpV1-dsRNAs triggers activation of type I IFN signaling in host cells. a** Upregulation of type I IFN and type I IFN-stimulated genes in IEC4.1 cells transfected with CSpV1-dsCA or CSpV1-dsRdRp for 24 h as revealed by RT-qPCR. **b** Slight activation of type I IFN signaling in IEC4.1 cells transfected with CSpV1-ssRdRp-F, CSpV1-ssRdRp-R, CSpV1-ssCA-F and CSpV1-ssCA-R, respectively, for 24 h. Data (in **a** and **b**) are from three biological replicates and presented as mean values ± SD. *p* values were determined by one-way ANOVA followed by Tukey's HSD test. **c** Heatmap of gene expression profiles in IEC4.1 cells following *C. parvum* infection or cells transfected with CSpV1-dsCA or CSpV1-dsRdRp or in combination. Cells were infected for 24 h or transfected with CSpV1-dsRNAs for 24 h. The heatmap representing differentially expressed genes (log10 fold changes) for each biological replicate (3 RNA-seq replicates each group). **d** Venn Diagrams demonstrating genes differentially expressed (with adjusted *p* values < 0.05) in IEC4.1 cells following infection (24 h p.i.) or transfected with CSpV1-dsCA or CSpV1-dsRdRp or their combination (for 24 h). **e** Heatmap representing expression of type I IFN-stimulated genes in infected IEC4.1 cells (24 h p.i.) or cells transfection of CSpV1-dsCA or CSpV1-dsRdRp or their combination (24 h). Data (in **c**–**e**) are from three RNA-Seq biological replicates for each group. **f** Interference with CSpV1-RNA delivery through transfection of host cells with siRNA combinations (siPOOLs) designed to target CSpV1-dsRNAs. IEC4.1 cells were transfected with the siPOOLs for 24 h, followed by infection for 24 h. Levels of CSpV1-dsRdRp were measured by RT-qPCR. A non-specific scrambled siRNA (siControl) was used for control. **g** Infection burden in IEC4.1 cells transfected with the siPOOLs or siControl for 24 h, followed by infection for 24 h. Levels of *cpHsp70* were measured by RT-qPCR. **h** Attenuation of *C. parvum*-triggered *Ifnb1* transcription in cells treated with siPOOLs. IEC4.1 cells were transfected with the siPOOLs or siControl for 24 h, followed by infection for 24 h, and *Ifnb1* levels were measured. Data (in **f**–**h**) are from three biological replicates and presented as mean values ± SD. *p* values were determined by two-way ANOVA followed by Tukey's HSD test (in **f** and **h**). Source data are provided as a Source Data file.

epithelial cells, resulting in detrimental effect on intestinal anti-*C. parvum* defense. It is well-documented that the production of type I IFNs could serve as a double-edged sword: type I IFNs provide early resistance against acute viral infections but are detrimental to the host during certain bacterial infections and chronic viral infections[8,42]. Moreover, these "miscellaneous upregulated genes" whose expression levels were further enhanced in infected *Villin.Ifnar1*[−/−] mice may also contribute to the resistance to *C. parvum* infection in the *Villin.Ifnar1*[−/−] neonatal mice. Whether proteins coding by these misc. upregulated genes can directly regulate intestinal epithelial cell anti-*C. parvum* defense, how the type I IFN signaling may modulate their expression in the intestine following infection, and whether the microbiota composition may play a role are important questions to be addressed.

Another key finding of this study is host delivery of CSpV1-dsRNAs and its association with type I IFN signaling activation in epithelial cells following *C. parvum* infection. Due to lack of the machinery for mammalian cell entry, CSpV1 is transmitted only by intracellular routes[20] and thus, it is unlikely that CSpV1 virus can directly infect intestinal epithelial cells. The predominant cytoplasmic localization of CSpV1-dsRNAs in infected cells, not in cells exposed to heat-treat parasites, indicates that direct parasite-host interaction is required for CSpV1-dsRNA delivery. CSpV1-dsRNAs in *C. parvum* are mainly localized to the sporozoite apical organelle region[21]. At the initial stage of host cell internalization, *C. parvum* sporozoites attach to the epithelial cell membrane of at their apical tip[28,30]. Subsequent discharge of apical organelle rhoptry and microneme contents into the host cell, including several parasite proteins such as ROP1 (cgd3_1770) and ROP3 (cgd3_1710), may contribute to parasite entry and parasitophorous vacuole formation[30,43,44]. Therefore, it is plausible that CSpV1-dsRNAs or CSpV1 virus particles are delivered into host cells in concert with parasite apical organelle discharge. Consequently, delivery of CSpV1-dsRNAs activates type I IFN signaling in infected cells through activating the Pkr and Rig-I/Mavs/Sting signal pathways. A comparable induction of type I IFN gene transcription was detected in cells transfected with CSpV1-dsRdRp or CSpV1-dsCA fragment, suggesting that CSpV1-dsRNAs trigger host type I IFN gene transcription in a sequence-independent manner. Of note, *C. parvum* infection triggers host type I IFN response through CSpV1-dsRNA-mediated activation of Pkr and Rig-I/Mavs/Sting signaling, in a TLR3-independent manner. TLR3 recognition accounts for intestinal type III produce[12] but is not required for type I IFN response in infected intestinal epithelial cells, suggesting distinct mechanisms to trigger type I and III signaling in host cells following *Cryptosporidium* infection.

CSpV1 is the only virus identified so far in various *Cryptosporidium* spp[20]. and whether CSpV1 has an impact on parasite pathogenesis is unclear. The dsRNA viruses of some other parasites can affect key aspects of parasite biology, including their virulence[45–47]. The presence of *Leishmania RNA virus 1* (LRV1) and *Trichomonas vaginalis* virus is strongly linked to parasite pathogenesis in human infections by *Leishmania guyanensis* and *Trichomonas vaginalis*[48,49]. LRV1 within the *Leishmania* parasite exacerbates the pathogenesis of human leishmaniasis[48,49], and the presence of the virus correlates with drug treatment failures[48]. Importantly, the fecundity of *C. parvum* appears to be correlated with its intracellular CSpV1 levels[50]. We speculate that high levels of type I IFN signaling in the intestinal epithelium induced by CSpV1-dsRNAs following *C. parvum* infection triggers the host's antiviral response preferentially over a strong antiparasitic response, which in turn benefits parasite survival. Accordingly, CSpV1 could be a new therapeutic target for cryptosporidiosis. A modest effect of anti-retroviral protease inhibitors, including indinavir which targets a HIV protease in HIV virus[51], on the development of cryptosporidiosis was previously reported[41,52,53]. We observed that ribavirin, antiviral reagent that interferes with the synthesis of viral mRNA[54], can also promote the inhibitory effect of antiparasitic drugs on *C. parvum* infection of intestinal epithelium. Although the underlying mechanisms are still unclear, our finding on the role of CSpV1 virus in attenuating host anti-parasitic defense may provide a mechanistical clue for the inhibitory effects of antiviral reagents on *Cryptosporidium* infection. Interestingly, the only FDA-approved anti-*C. parvum* drug nitazoxanide also displays some effects in the treatment of chronic hepatitis B and C[55,56].

To conclude, our findings in this study indicate a detrimental effect of type I IFN signaling in intestinal epithelial cells on intestinal anti-*C. parvum* defense and reveal an important strategy of immune evasion by the parasite through the delivery of CSpV1-dsRNAs. Better understanding the functional impact of CSpV1-mediated type I IFN responses on mucosal anti-*C. parvum* immunity would provide mechanistic insights into host-*C. parvum* interactions, relevant to future development of novel therapeutic strategies.

## Methods

### Ethics statement

All research studies involving the use of animals were reviewed and approved by the Institutional Animal Care and Use Committees of the Creighton University School of Medicine at Omaha, Nebraska and were carried out in strict accordance with the recommendations in the Guide for the Care and Use of Laboratory Animals. The 3D enteroids are derived from human intestinal tissues from an existing human tissue bank (IRB protocol #714-18; University of Nebraska Medical Center, NE, USA).

### *C. parvum*, cell lines, and human intestinal epithelial monolayers

*C. parvum* oocysts were harvested from calves inoculated with the Iowa strain originally obtained from Dr. Harley Moon at the National Animal Disease Center (Ames, IA, USA) and purchased from a commercial source (Bunch Grass Farms, Deary, ID, USA). For in vitro, ex vivo, and in vivo infection, oocysts were treated with 1% sodium hypochlorite (Spectrum Chemical, catalogue no, S1669) on ice for

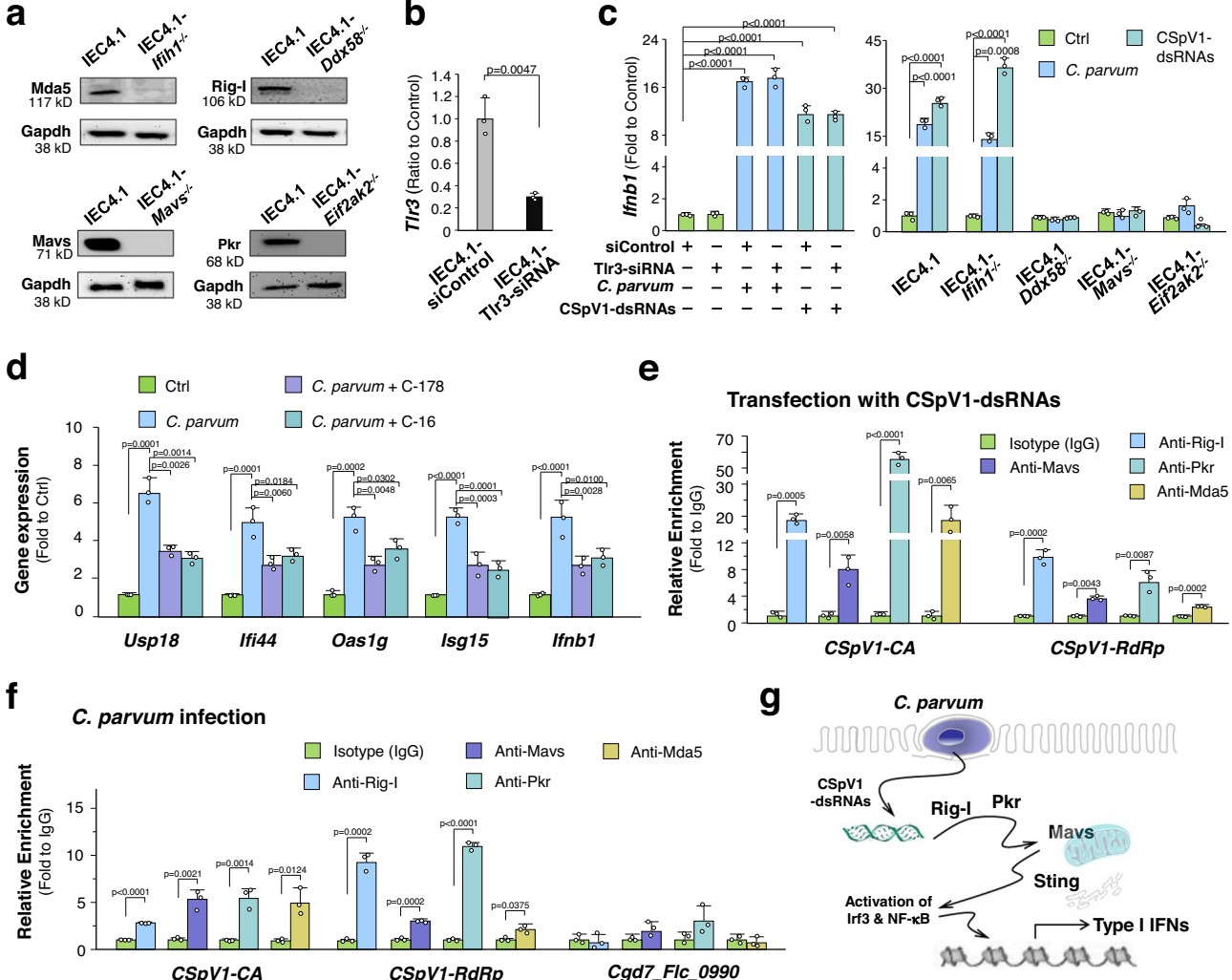

**Fig. 5 | CSpV1-dsRNAs trigger type I IFN signaling through activation of the Pkr and Rig-I/Mavs/Sting signaling pathways. a** Stable IEC4.1 cells deficient in *Ifih1*, *Ddx58*, *Mavs*, or *Eif2ak2* generated using the CRISPR/Cas9 approach as confirmed by Western blot. Representative gel images from three independent experiments are shown. **b** Knockdown of Tlr3 with an siRNA in IEC4.1 cells as confirmed by RT-qPCR. Data are from three biological replicates and presented as mean values ± SD. *p* values were determined by two-tailed unpaired Student's *t*-test. **c** *C. parvum*- and CSpV1-dsRNA-induced *Ifnb1* expression in IEC4.1 cells deficient in *Ifih1*, *Ddx58*, *Mavs*, or *Eif2ak2* and in cells treated with the Tlr3-siRNA. Cells were exposed to *C. parvum* or transfected with CSpV1-dsRNAs for 24 h. To knockdown Tlr3, cells were first treated with the Tlr3-siRNA for 24 h and then exposed to infection or CSpV1-dsRNA transfection. *Ifnb1* expression was measured by RT-qPCR. **d** Inhibition of Sting signaling and Pkr signaling on *C. parvum*-induced expression of type I IFNs and type I IFN-controlled genes. IEC4.1 cells were exposed to *C. parvum* infection for 8 h and cultured for additional 16 h in the presence or absence of the inhibitor

C-178 (to Sting signaling) or C-16 (to Pkr signaling). *Usp18, Ifi44, Oas1g, Isg15, and Ifnb1* RNA levels were measured by RT-qPCR. *p* values were determined by one-way or two-way ANOVA followed by Tukey's HSD test (in **c**, **d**). **e**, **f** Interaction between CSpV1-dsRNAs and components in the Rig-I/Mavs/Pkr pathways in IEC4.1 cells following infection or CSpV1-dsRNA transfection. Cells were exposed to *C. parvum* or transfected with CSpV1-dsRNAs for 24 h. Cell lysates were collected for RNA-protein interaction measurement using RNA immunoprecipitation assay. For infected cells, a parasite RNA, *Cgd7_Flc_0990*, was used for control. *p* values were determined by two-tailed unpaired Student's *t*-test (in **e**, **f**). Data (in **c**–**f**) are from three biological replicates and presented as mean values ± SD. **g** Model of CSpV1-dsRNAs to trigger type I IFN signaling in host cells. CSpV1-dsRNAs can be recognized by RIG-I and PKR in infected cells, resulting in activation of the Pkr and Rig-I/Mdv5/Sting signaling pathways and subsequently, activation of type I IFN signaling. Source data are provided as a Source Data file.

20 min and washed twice with 1x phosphate-buffered saline (PBS, IBI Scientific, catalogue no. IB70165) followed by washing twice with DMEM/F12 Medium (ATCC, catalogue no. 30-2006).

The neonatal murine intestinal epithelial cell line (IEC4.1) was a kind gift from Dr. Pingchang Yang (McMaster University, Hamilton, Canada). The human intestinal adenocarcinoma HCT-8 cells were obtained from ATCC (CCL-244) and cultured in RPMI-1640 Medium (ATCC, catalogue no. 30-2001) supplied with 10% FBS (Gibco, catalogue no. 10082147) and 100 U/ml penicillin-streptomycin (Gibco, catalogue no. 15140-122). The human 2D intestinal epithelial monolayers were generated from 3D enteroids derived from human intestinal tissues from an existing human tissue bank (IRB protocol #714-18;

University of Nebraska Medical Center, NE, USA). Intestinal villus/crypt components were isolated and cultured using an approach modified from our previous studies in mice[57]. Briefly, human biopsies were washed once with ice-cold Dulbecco's phosphate buffered saline without $Ca^{2+}$ and $Mg^{2+}$ (DPBS, Gibco, catalogue no. 14190144) and twice with ice-cold chelation buffer [1x DPBS, 2% sorbitol (Fisher Scientific, catalogue no. BP439-500), 1% sucrose (Fisher Scientific, catalogue no. BP220-1), 1% bovine serum albumin (BSA, Thermo Scientific, catalogue no. 37525)]. Washed biopsies were immersed in freshly prepared ice-cold chelation buffer with 2 mM EDTA (Sigma-Aldrich, catalogue no. E9884), shaken gently for 30 min at 4 °C on a Pulsing Vortex Mixer (Fisher Scientific, catalogue no. 02-215-422) and then vigorously until it

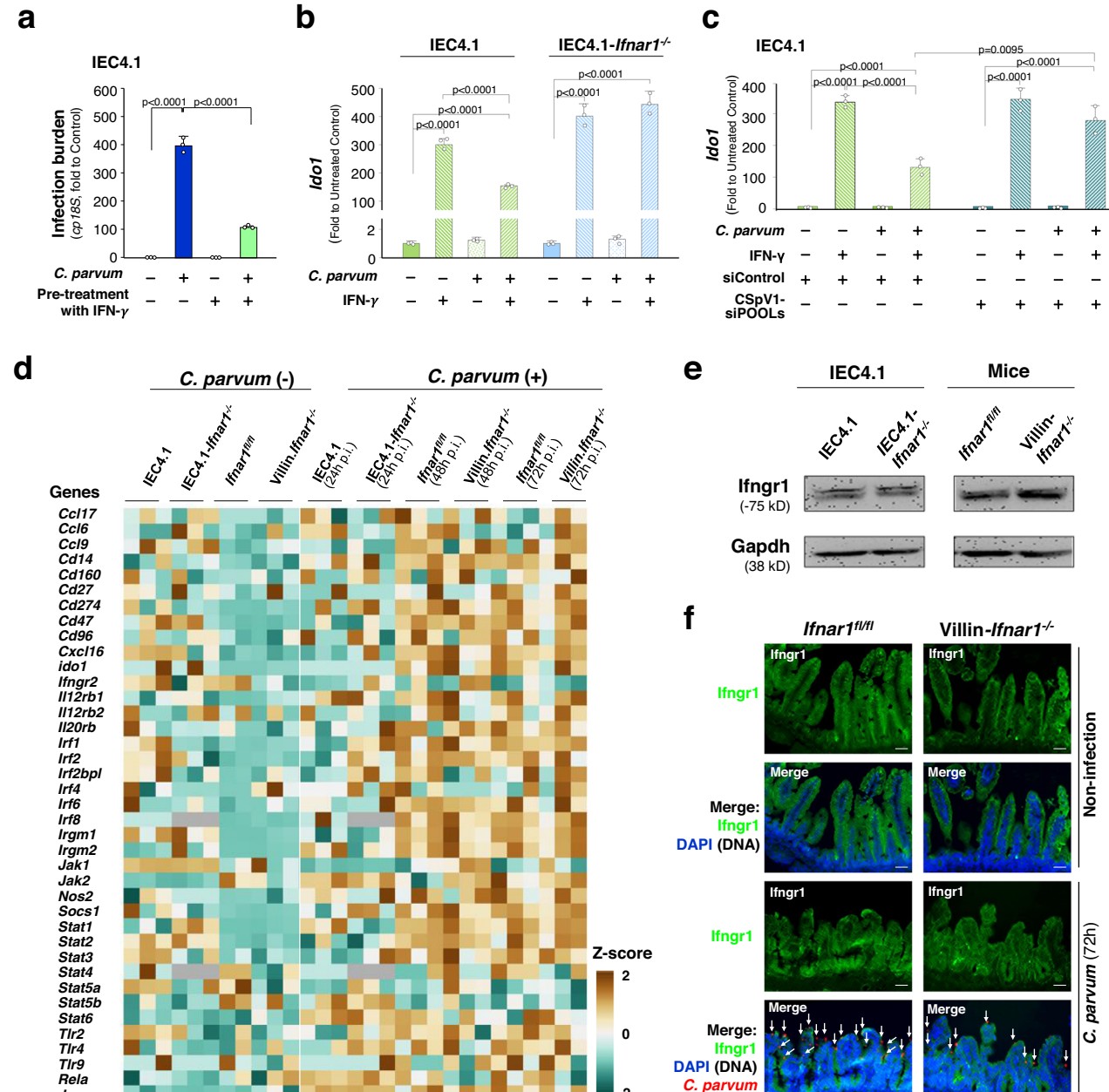

**Fig. 6 | Impaired cellular response to IFN-γ stimulation in infected epithelial cells and its association with *C. parvum*-induced type I IFN signaling activation.** **a** Pre-treatment of IEC4.1 cells with IFN-γ decreased *C. parvum* infection burden. Cells were first treated with IFN-γ (10 ng/ml) for 8 h, exposed to *C. parvum* for 24 h, and *cp18S* was quantified by RT-qPCR. **b** Impaired cellular response to IFN-γ stimulation in infected IEC4.1 cells associated with type I IFN signaling. IEC4.1 and *Ifnar1*−/− IEC4.1 cells were first exposed to *C. parvum* infection for 24 h followed by IFN-γ stimulation for additional 2 h. Cellular response to IFN-γ stimulation was reflected by the expression level of the *Ido1* gene. **c** Knockdown of CSpV1-dsRNA delivery in infected IEC4.1 cells through transfection of the specific siRNA combinations (siPOOLs) partially restored the induction of *Ido1* gene expression in response to IFN-γ stimulation. A non-specific control siRNA (siControl) was used for control. Data (in **a**–**c**) are from three biological replicates and presented as mean values ± SD. *p* values were determined by two-way ANOVA followed by Tukey's HSD test (in **a**–**c**). **d** Heatmaps representing altered expression levels of selected genes

for the key components associated with IFN-γ signaling in infected IEC4.1 cells or neonatal intestinal epithelium. IEC4.1 cells (24 h p.i.) or intestinal epithelium from *Ifnar1*fl/fl and *Villin.Ifnar1*−/− neonates (5 days old) after infection (48 h and 72 h p.i.) were collected, followed by RNA-Seq. Expression levels of genes for key components associated with IFN-γ signaling are shown. Data are from three RNA-Seq biological replicates for each group. **e** Protein content of the Ifngr1 subunit of IFN-γ receptor in IEC4.1 and *Ifnar1*−/− IEC4.1 cells, intestinal epithelium from *Ifnar1*fl/fl and *Villin.Ifnar1*−/− neonates as assessed by Western. Gapdh was used as a loading control. Representative gels from three independent experiments are shown. **f** Intestinal epithelium of *Ifnar1*fl/fl and *Villin.Ifnar1*−/− neonatal mice (either uninfected or 72h-infected animals) showed a similar staining for Ifngr1 protein by immunofluorescence. Representative images from 4 independent experiments are shown. Blue: DAPI (DNA), green: Ifngr1, red: *C. parvum* (arrows). Bars = 50 μm. Source data are provided as a Source Data file.

became mostly opaque with dislodged crypt and villus particles. The crypt pellets were collected after quick centrifugation for 30 s and washed in ice-cold chelation buffer for three times. The pellets were then resuspended in ice-cold basement membrane matrix comprising

complete human minigut 3D medium and Matrigel Membrane Matrix (Fisher Scientific, catalogue no. CB-40234) in 1:1 ratio v/v, and plated in 37 °C pre-warmed 35 mm dishes (30 μL mix per spot, 12 spots) or 60 mm dishes (30 spots). The basement membrane matrix was

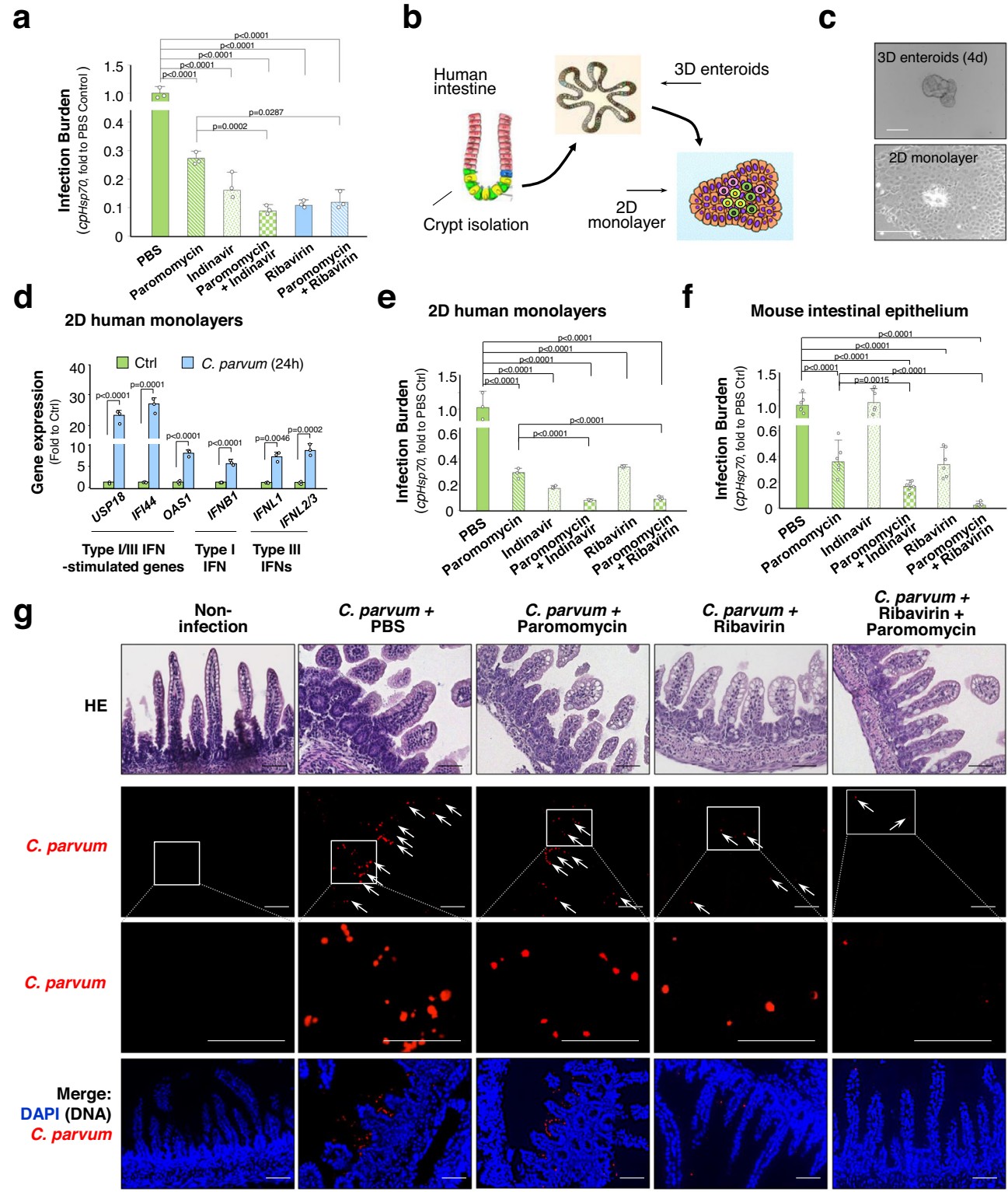

overlaid with complete human minigut 3D medium after incubation in 37 °C, 5% $CO_2$ for 20 min. Complete human minigut 3D medium was prepared in a base solution of Wnt-3A conditioned medium and human minigut basic medium in 1:1 ratio v/v with 1 μg/ml hR-spondin1 (Peprotech, catalogue no. 120-38), 100 ng/ml hNoggin (Biotech, catalogue no. 6057-NG), 50 ng/ml hEGF (Sigma-Aldrich, catalogue no. E9644), 500 nM A-83-01 (Tocris Bioscience, catalogue no. 2939), 10 nM [Leu]15-Gastrin1 (Sigma-Aldrich, catalogue no. G9145), 10 nM Nicotinamide (Sigma-Aldrich, catalogue no. N0636), and 1 mM N-Acetylcysteine (Sigma-Aldrich, catalogue no. A9165). Wnt-3A

conditioned medium was generated in house with L-Wnt3a cell line (ATCC, catalog no. CRL-2647) following the manufacturer's instructions. Human minigut basic medium comprises 1x Advanced DMEM/ F12 Medium (Gibco, catalogue no. 12634-010), 1x glutamine (Gibco, catalogue no. 35050-061), 100 U/ml penicillin-streptomycin, 1x N2 supplement (Gibco, catalogue no. 17502-048), 1x B27 supplement (Gibco, catalogue no. 17504-044), and 1% BSA. All above steps were carried out in buffers kept on ice. Complete human minigut 3D culture medium was changed every other day. For enteroid propagation, complete human minigut 3D medium should be freshly prepared.

**Fig. 7 | Effects of anti-viral and anti-parasitic drugs on *C. parvum* infection of intestinal epithelium. a** Effects of anti-viral and anti-parasitic drugs on *C. parvum* infection of cultured HCT-8 cells. Cells were exposed to *C. parvum* infection for 24 h in the presence or absence of anti-viral and anti-parasitic drugs or their combination, followed by RT-qPCR of *cpHsp70*. **b** Schematic diagram of human 3D enteroids and 2D intestinal epithelial monolayer cultures. **c** Brightfield microscopy images of 3D enteroids (4 days culture from isolated human crypts) and 2D monolayers derived from 3D enteroids are shown. Representative images from three independent experiments are shown. Bars = 50 μm. **d** Expression of type I/III IFN and type I/III IFN-stimulated genes in human 2D intestinal epithelial monolayers following infection. Human 2D intestinal epithelial monolayer cultures were exposed to *C. parvum* infection for 24 h. Expression levels of *USP18*, *IFI44*, *OAS1*, *IFNB1*, *IFNL1*, and *IFNL2/3* were measured by RT-qPCR. **e** Effects of anti-viral and anti-parasitic drugs on *C. parvum* infection of human 2D intestinal epithelial

monolayers. 2D monolayers were exposed to *C. parvum* infection for 24 h, in the presence or absence of anti-viral and anti-parasitic drugs or their combination, followed by RT-qPCR measurement of *cpHsp70*. **f, g** Effects of antiviral and anti-parasitic drugs on *C. parvum* intestinal infection in neonates. Five-day-old C57BL/6 neonates were orally administered *C. parvum* oocysts; antiviral and antiparasitic drugs or their combination were given at 2 h p.i. followed by daily for 3 days. Intestinal tissues were collected and infection burden was measured by RT-qPCR of *cpHsp70* (**f**) and immunofluorescent staining of *C. parvum* (**g**). Blue: DAPI (DNA), red: *C. parvum* (arrows). A higher magnification of the boxed region to visualize *C. parvum* is shown. Bars = 50 μm. Data represent three biological replicates (**a**, **d**, **e**) or 6 mice in each group (**f**, **g**) and presented as mean values ± SD. *p* values were determined by one-way ANOVA followed by Tukey's HSD test (in **a**, **e**, **f**) or by two-tailed unpaired Student's *t*-test (in **d**). Source data are provided as a Source Data file.

Monolayer cultures were derived from 3D enteroids as previously described[16,58].

### Generation of conditional intestinal epithelial *Ifnar1*$^{-/-}$ mice

The conditional intestinal epithelial type I IFN receptor *Ifnar1* KO mice (*Villin.Ifnar1*$^{-/-}$ mice) were generated from the cross-breeding of the *Ifnar1*$^{fl/fl}$ mice (JAX, catalogue no. 028256, which carry a floxed allele with *loxP* sites flanking exon 3 of the type-I-IFN-α/β receptor gene) and the *Vil1-cre* mice (JAX, catalogue no. 004586, which contain the villin1 promotor directing expression of mouse Cre recombinase in villus and crypt epithelial cells of the intestine). Mice were housed and bred in the animal care facility under a 12/12 h light/dark cycle, 20 °C ambient temperature with humidity around 50%. Both breeding pairs of both strains were purchased from the Jackson Laboratory (Bar Harbor, ME) and cross-bred according to the manufacturer's guide. The genotypes of littermates were identified using mouse tail DNA for PCR using primers designed from the Jackson Laboratory: wild type (loxP 395 bp and Cre$^+$, M1; loxP 325 bp and Cre$^-$, M2), heterozygous (loxP 395/325 bp and Cre$^+$, M3) and homozygous (loxP 325 bp and Cre$^+$, M4). Lack of *Ifnar1* exon3 expression in the intestinal epithelium of *Villin.Ifnar1*$^{-/-}$ mice was further confirmed by RT-qPCR (Supplementary Fig. 1a).

### Reagents, infection models, antiparasitic and antiviral agents, and infection assays

C-178 was purchased from ProbeChem (catalogue no. PC-35310) and C-16 from Sigma-Aldrich (catalogue no. 527451). Models of intestinal cryptosporidiosis using intestinal epithelial cell lines and 2D epithelial monolayers were employed as previously described[16,57,58]. Infection was done in culture medium (DMEM/F12 with 100 U/ml penicillin-streptomycin) containing viable *C. parvum* oocysts (oocysts with host cells in a 1:4 ratio). A well-developed infection model of cryptosporidiosis in neonatal mice was used for in vivo experiments[59,60]. Briefly, neonates (5 days old, both male and female animals were randomly used) were received *C. parvum* oocysts by oral gavage ($10^6$ oocysts per animal). Mice receiving 1× PBS by oral gavage were used as control. For animals treated with antiparasitic and antiviral reagents, first treatment was administered 2 h after *C. parvum* infection, followed by daily for 3 days. Animals received antiparasitic agent (paromomycin, Sigma-Aldrich, catalogue no. P9297, 25 mg/kg body weight/day, P.O.) or antiviral agents (ribavirin, Sigma-Aldrich, catalogue no. R9644, 40 mg/kg body weight/day, P.O., or indinavir, Sigma-Aldrich, catalogue no. SML0189, 24 mg/kg body weight/day, I. P.) or a combination of antiparasitic and antiviral agents (paromomycin, 25 mg/kg body weight/day, with ribavirin, 40 mg/kg body weight/day; or paromomycin, 25 mg/kg body weight/day, with indinavir, 25 mg/kg body weight/day). At various time points after *Cryptosporidium* or vehicle/drug administration, ileum intestinal tissues (4 cm of small intestine tissue from the ileocecal junction) were collected from the animals (minimum of 6 mice from each group). Biochemical analysis, RT-qPCR, immunofluorescent staining, and histology were performed as previously

reported[57,58]. For in vitro and ex vivo infection, cell cultures or 2D monolayers were exposed to *C. parvum* infection for 4 h and culture medium was changed to remove free parasite. Paromomycin (500 μg/ml), indinavir (200 μg/ml), ribavirin (20 μg/ml) or a combination of antiparasitic and antiviral agents was then added to the medium and cultured for additional 20 h (a total 24 h p.i.) followed by infection burden measurement. Infection burden was quantified by measuring levels of *C. parvum* 18 s (*cp18s*) or Hsp70 (*cpHsp70*) gene[13]. Immunofluorescent staining of *C. parvum* was carried out using the rabbit antiserum against *C. parvum* membrane proteins as previously described[61,62]. For parasite counting, the number of intracellular parasites in each intestinal section following immunofluorescent staining of *C. parvum* was determined from 20 randomly chosen fields at 60X total magnification and presented as average parasite number per 60X field. Other antibodies used for immunofluorescent staining include anti-PCNA antibody (Cell Signaling, catalogue no. 13110, 1:1000), anti-EpCAM antibody (Abcam, catalogue no. ab71916, 1:200), anti-Lysozyme antibody (Abcam, catalogue no. ab108508, 1:500), anti-Villin antibody (Santa Cruz, catalogue no. sc-58897, 1:200), anti-mouse-Cruz fluor 488-conjugated (Santa Cruz, catalogue no. sc-516176, 1:200), anti-rabbit-Cruz fluor 488-conjugated (Santa Cruz, catalogue no. sc-516248, 1:200), anti-bovine Alexa fluor 594-conjugated (Jackson Immuno Research lab, catalogue no. 101-585-003, 1:250).

### RT-qPCR

For quantitative analysis of RNA expression, comparative real-time PCR was performed as previously reported[57,58], using the iTaq™ Universal SYBR® Green Supermix (Bio-rad, Hercules, CA, USA, catalogue no. 1725124). Briefly, total RNA was isolated using TRIzol™ Reagent (ThermoFisher Scientific, catalogue no. 15596018) and possible remaining DNA was removed with the DNA-*free*™ DNA Removal Kit (ThermoFisher Scientific, catalogue no. AM1906). The amount of RNA was measured by quantitative, strand-specific RT-PCR using the iCycler iQ Real-time detection system (Bio-Rad CFX Manager v3.1) with 25 ng of template cDNA from reverse transcription for each RNA gene of interest. The expression level of each RNA was calculated using the $^{\Delta\Delta}$Ct method and normalized to *Gapdh*. All sequences of PCR primers are listed in Supplementary Data 4.

### siRNAs, CRISPR/Cas9 and stable KO IEC4.1 cells

Custom-designed RNA oligos against CSpV1-dsRNAs and scrambled siRNA-control were synthesized by Integrated DNA Technologies (Coralville, Iowa) and siRNA to Tlr3 was from Santa Cruz (sc-40259). Sequences of siRNAs are listed in Supplementary Data 4. siRNAs were transfected into cells with Lipofectamine™ RNAiMAX Transfection Reagent according to the manufacturer's protocol (ThermoFisher Scientific, catalogue no. 13778150). Stable IEC4.1 cell lines with deficiency in *Ifnar1*, *Ddx58*, *Eif2ak2*, *Mavs* or *Ifih1* genes were generated through transfection of cells with the CRISPR/Cas9 KO and the HDR

plasmid, respectively, according to the manufacturer's instruction and as previously described[16]. Briefly, the gene-specific CRISPR/Cas9 KO plasmids and its corresponding HDR plasmids were co-transfected into the cells using the UltraCruz® Transfection Reagent (Santa Cruz, catalogue no. sc-395739) and plasmid transfection medium (Santa Cruz, catalogue no. sc-108062), followed by selection with puromycin (5 μg/ml, Santa Cruz, catalogue no. sc-108071). The IFN-α/βRα CRISPR Plasmids(m) (sc-421047), Rig-I CRISPR Plasmids(m) (sc-432915), PKR CRISPR Plasmids(m) (sc-422410), Mavs CRISPR Plasmids(m) (sc-432790), Mda5 CRISPR Plasmids(m) (sc-427977) and their corresponding HDR plasmids were obtained from Santa Cruz Biotechnology (Dallas, TX, USA). The primers used to screen the CRISPR/Cas9 KO cell lines are listed in Supplementary Data 4.

## Cytoplasmic extracts, Western blot, and dot blot

Cells were grown to 80% confluence, treated, and collected at designed time points. Cells were washed with 1× PBS and detached with 0.05 M EDTA in 1× PBS. Detached cells were pelleted by centrifugation at $100 \times g$ for 3 min. Cytoplasmic extracts were prepared using the standard approach[32]. Briefly, pelleted cells were resuspended in lysis buffer [10 mM Tris-HCl (Fisher Scientific, catalogue no. BP-152-1), pH 7.4, 10 mM NaCl (Fisher Scientific, catalogue no. S271-3), 3 mM MgCl$_2$ (MP Biomedicals, Irvine, CA, catalogue no. 191421), 100 U/ml RNase inhibitor (Thermo Fisher Scientific, catalogue no. AM2696)] and incubated on ice for 10 min. Cell lysates were centrifuged at 100 g for 2 min at 4 °C and the supernatants saved as cytoplasmic extract.

Western blot and dot blot were performed as reported in our previous study[57,58]. Briefly, cell lysates were collected and SDS-PAGE gel electrophoresis was used to separate denatured proteins on 10% TGS gels. Proteins were transferred to Immun-Blot® PVDF Membrane (Bio-Rad, catalogue no. 1620177). The membranes were blocked with Blotting Grade Blocker Non Fat Dry Milk (Bio-Rad, catalogue no. 1706404XTU) for 1 h and then incubated with the primary antibody. After washing with 1× Tris Buffered Saline with 0.1% Tween® 20 for three times (10 min each), secondary antibody was applied. For dot blot, RNA was spotted on the Amersham Hybond-N+ membrane (Avantor, catalogue no. 95038-336) followed by UV crosslinking and antibody incubation. Membranes were washed and developed using Clarity Western ECL Substrate (Bio-Rad, catalogue no. 1705061) following the manufacturer's instructions. Blot imaging was obtained and analyzed using the AlphaEase FC Software v6.0.2. The following antibodies were used for blotting: anti-Isg15 (Santa Cruz, catalogue no. sc-166755, 1:100), anti-Usp18 (Biomatik, catalogue no. CAC08089, 1:500), anti-Ifngr1 (Thermo Fisher Scientific, catalogue no. MA5-35147, 1:500), anti-Mda5 (Abcam, catalogue no. Ab79055, 1:50), anti-Rig-I (Proteintech, catalogue no. 20566-1-AP, 1:500), anti-Mavs (Proteintech, catalogue no. 14341-1-AP, 1:2000), anti-Pkr (Santa Cruz, catalogue no. sc-6282 AC, 1:200), anti-viral dsRNAs (J2) (Millipore, catalogue no. MABE1134, 1:60), anti-Gapdh (Santa Cruz, catalogue no. sc-32233, 1:1000), anti-rabbit-HRP-conjugated secondary antibody (Santa Cruz, catalogue no. sc-2357, 1:5000), anti-mouse-HRP-conjugated secondary antibody (Santa Cruz, catalogue no. sc-516102, 1:5000), and anti-viral dsRNA (J2) (Millipore, catalogue no. MABE1134, 1:60).

## RNA-Seq and bioinformatics analysis

Total RNAs were extracted using Rneasy Mini Kit (QIAGEN, catalogue no. 74104). Transcriptome sequencing and result processing were performed by the BGI Americas Corporation (Cambridge, MA) with DNBSEQ™ sequencing technology platforms as previously reported[40]. The quality of RNA was evaluated with a 2100 Bioanalyzer (Agilent Technologies, USA) and Samples classified as good RNA quality had a similar RNA pattern at visual inspection in the Bioanalyzer, an RNA Integrity Number (RIN) > 7 and a concentration >20 ng/ul. Total RNA was fragmented into short fragments and mRNA was enriched using oligo (dT) magnetic beads, followed by cDNA synthesis. Double-stranded cDNA was purified and enriched by PCR amplification, after which the library products were sequenced using DNBSEQ-500 platform with read length of PE100, averagely generated about 4.94 Gb bases per sample. Low quality reads (more than 20% of the bases qualities are lower than 10) and reads with adaptors and reads with unknown bases (N bases more than 5%) were filtered to get the clean reads using the internal software SOAPnuke (version: v1.5.2, parameters: -l15-q0.5-n0.1) in the DNBSEQ-500 platform. After filtering, the clean reads were stored in FASTQ format and read alignments were performed by HISAT2 (version: v2.0.4; parameters: -phred64-sensitive-no-discordant-no-mixed-I 1-X 1000) and Bowtie2 (version: v2.2.5) referring to the *Mus_musculus* genome (*GCF_000001635.26_GRCm 38.p6*) from NCBI database. Raw RNA-seq reads were then filtered out the unreliable sites to remove low-quality reads using the GATK program for each sample. Final gene reads were normalized to RPKM (reads per kilobase of transcript, per Million mapped reads) as relative gene expression levels.

Differential expression analysis was performed by the BGI, using the Dr. TOM data system. Altered (upregulated or downregulated) expression of genes was expressed as log2FC or log10FC, which represents log-transformed fold change (log2FC = log2[B] − log2[A] or log10FC = log10[B] − log10[A], while A and B represent values of gene expression from different treatment conditions). Statistical significance was determined based on two-tailed Wald test followed by Benjamini−Hochberg procedure for multiple comparison correction[63]. The Venny 2.1 software was used to identify genes common to the different treated groups[64]. Gene set enrichment analysis (GSEA) was run using pre-ranked mode for all genes by R package "clusterProfiler" based on log2FC derived from the differential expression analysis by weighted Kolmogorov−Smirnov test and $p$ value was adjusted by Benjamini−Hochberg method[65]. Visualization of GSEA results were achieved by R package "ggplot2"[66]. Volcano plot was generated using R package "ggplot2" with log2FC and adjusted $p$ value derived from differential expression analysis. Heatmaps were generated using R package "ComplexHeatmap"[67]. Expression z-score was calculated based on the relative gene expression levels[68].

## Synthesis of CSpV1-dsRNAs and transfection

*C. parvum* sporozoite total RNA was isolated using TRIzol (Invitrogen, catalogue no. 15596-026). CSpV1-dsRNAs were reverse-transcribed to cDNAs using MMLV-RT (Thermo Fisher Scientific, catalogue no. 28025-013) with sporozoite total RNA as template and specific primers for CSpV1dsRNA1_RdRpFull_R1 (5′-GGCCGATCATAGCTACT CC-3′), CSpV1dsRNA2_CAFull_R1 (5′-GGCCGATCATAGCTACTTTCATA AG-3′). Amplification of full length CSpV1-RdRp and CSpV1-CA was performed using the Platinum™ *Taq* DNA Polymerase High Fidelity system (Thermo Fisher Scientific, catalogue no. 11304011) with reverse-transcribed cDNA templates and specific primers CSp V1dsRNA1_RdRpFull_F1 (5′-GGAAAGAAGCATAGCTCAATTCTC-3′), CSp V1dsRNA1_RdRpFull_R1 (5′-GGCCGATCATAGCTACTCC-3′), CSpV1ds RNA2_CAFull_F1 (5′-GGAAAAGAGGAACAGCAACTC-3′), CSpV1dsRNA2_ CAFull_R1 (5′-GGCCGATCATAGCTACTTTCATAAG-3′). Amplified products were sent to Genewiz (South Plainfield, NJ) for Sanger Sequencing. Plasmid constructs for CSpV1-dsRdRp in vitro transcription was generated by cloning PCR-amplified full length CSpV1-RdRp into the pSPT18 vector (Sigma-Aldrich, catalogue no. 10999644001), into the HindIII and KpnI sites, in the orientation for SP6 transcription. PCR-amplified full length CSpV1-CA was A-tailed with *Taq* DNA Polymerase (NEB, catalogue no. M0267) and cloned into the pTargeT™ vector using the pTargeT™ Mammalian Expression Vector System (Promega, catalogue no. A1410). Single-strand CSpV1-RNAs were synthesized using SP6/T7 Transcription Kit (Sigma-Aldrich, catalogue no. 10999644001) with the pSPT18-CSpV1-RdRp and pTargeT™-CSpV1-CA plasmid constructs as templates. Sense and anti-sense CSpV1-ssRNAs were annealed to CSpV1-dsRdRp and CSpV1-dsCA respectively by

95 °C incubation for 5 min and gradual cooling down over 2 h. CSpV1-ssRNAs and CSpV1-dsRNAs were transfected to IEC4.1 cells with Lipofectamine 2000 (Thermo Fisher Scientific, catalogue no. 11668019) for 24 h. A non-specific scrambled dsRNA was transfected at the same time in control group. For specific details of the specific primers and RNA sequences, see the Supplementary Data 4.

## Fluorescent in situ hybridization

DIG-labeled antisense CSpV1-dsRNAs were synthesized employing SP6/T7 Transcription Kit (Sigma-Aldrich, catalogue no. 10999644001) as described above. Cells were incubated and treated on Nunc™ Lab-Tek™ II Chamber Slide™ System (Thermo Scientific, catalogue no. 154534). The cells were fixed in 4% paraformaldehyde (Thermo Fisher Scientific, catalogue no. 28908) with 5% acetic acid (Fisher Scientific, catalogue no. BP2401-212) and 0.9% NaCl for 30 min, and washed twice with diethylpyrocarbonate (DEPC, Sigma-Aldrich, catalogue no. 06756) treated 1× PBS. Fixed cells were permeabilized in 0.1% Triton X-100/PBS (Bio-rad, catalogue no. 1610407) with 100 U/ml RNase inhibitor for 10 min and washed three times with 1× DEPC-PBS. Hybridization occurred in DIG Easy Hyb Granules (Sigma-Aldrich, catalogue no. 11 796 895 001) according to manufacturer's guide. Blocking of hybridization was performed using Blocking Reagent (Sigma-Aldrich, catalogue no. 11 096 176 001). Subsequent immunofluorescence staining was performed with Anti-Digoxigenin-Fluorescein (Sigma-Aldrich, catalogue no. 11 207 741 910) in a concentration of 1:200 in 0.5% BSA/PBS for 1 h. The slides were washed twice with 1x DEPC-PBS and then mounted using Antifade Mounting Medium with DAPI (Vector, catalogue no. H-1200).

## RIP assay

The formaldehyde crosslinking RIP was performed as described[16]. Briefly, cells were treated with 0.3% Formaldehyde (Fisher Scientific, catalogue no. BP531-500) at 37 °C for 10 min. Crosslinking reactions were quenched by incubation with 0.25 M glycine (IBI Scientific, catalogue no. IB70194) at room temperature for 5 min. Cells were harvested by centrifugation at $800 \times g$ for 4 min at 4 °C and washed twice with ice-cold 1× PBS. Cell pellets were resuspended in whole cell extract buffer [20 mM HEPES pH 7.4 (Teknova, catalogue no. H1030), 0.2 M NaCl, 0.5% Triton X-100, 10% glycerol (Sigma-Aldrich, catalogue no. G7757), 1 mM EDTA, 1 mM EGTA (Sigma-Aldrich, catalogue no. E3889), 1 mM DTT (Fisher Scientific, catalogue no. BP1725), cocktail protease inhibitor (Sigma-Aldrich, catalogue no. 11697498001) and 100 U/ml RNase inhibitor]. The crosslinked complexes in cell lysates were solubilized by three rounds of sonication, 20 s each, in a Microson XL2007 ultrasonic homogenizer with a microprobe at an amplitude setting of 7 (output, 8–9 W). Insoluble materials were removed by microcentrifugation at $16,100 \times g$ for 20 min at 4 °C. Lysates were precleaned with 1× PBS washed Magna CHIP Protein A + G Magnetic Beads (Sigma-Aldrich, catalogue no. 16-663) and isotype IgG. The precleaned lysate was then diluted with whole cell extract buffer in 1:1 ratio v/v, mixed with the specific antibody- or isotype IgG-coated beads, and incubated with rotation at 4 °C overnight, followed by two washes with whole cell extract buffer. The collected immunoprecipitated RNP complexes and input were digested in RNA PK Buffer pH 7.0 [100 mM NaCl, 10 mM Tris-HCl pH 7.0, 1 mM EDTA, 0.5% SDS (Fisher Scientific, catalogue no. BP166500) and 100 U/ml RNase inhibitor] with 100 µg/ml Proteinase K (Thermo Scientific, catalogue no. E0491)] and incubated at 50 °C for 45 min with end-to-end shaking. Formaldehyde cross-links were reversed by incubation at 65 °C with end-to-end shaking for 4 h. RNA was extracted using the TRIzol™ Reagent and treated with DNA-free™ DNA Removal Kit. The presence of RNA was measured by RT-qPCR. Gene-specific PCR primer pairs are listed in Supplementary Data 4. The following antibodies were used: anti-Mda5 (ThermoFisher Scientific, catalogue no. 21775-1-AP, 4 µg/ml), anti-Rig-I (Proteintech, catalogue no. 20566-1-AP, 4 µg/ml), anti-Mavs (Proteintech, catalogue no. 14341-1-AP, 4 µg/ml), anti-Pkr (Proteintech, catalogue no. 18244-1-AP, 4 µg/ml), anti-viral dsRNAs (J2) (Millipore, catalogue no. MABE1134, 4 µg/ml), anti-Mouse-IgG (Cell Signaling, catalogue no. 5415 S), anti-Rabbit-IgG (Cell Signaling, catalogue no. 3900S, 4 µg/ml).

## ELISA

IEC4.1 cells were grown on 24 well plates to 80% confluency and were infected with *C. parvum* oocysts. At the indicated timepoint post infection, supernatants were collected and selected IFN protein levels were measured by mouse IFN-beta DuoSet ELISA (DY8234, R&D Systems), mouse IL28B/IFN-lambda 2/3 DuoSet ELISA (DY1789B, R&D Systems), and mouse IFN-gamma DuoSet ELISA (DY485, R&D Systems) according to the manufacturer's instructions. Briefly, 96-well microplate was coated with Capture Antibody and incubated overnight at room temperature after sealing. Samples or standards were added to the processed microplate and incubated at room temperature for 2 h after sealing. After washing, the detection antibody was added to the microplate and incubated at room temperature for 2 h. Streptavidin-HRP was added and incubated at room temperature for 20 min. After washing, the plate was incubated with the Substrate Solution at room temperature for 20 min. Stop solution was then added and a microplate reader was used to determine the optical density of each well immediately.

## Statistical and reproducibility

Statistical analysis was performed using GraphPad Prism 5 (GraphPad Software). All values are given as mean ± S.D. Means of groups were from at least three independent experiments or biological replicates. When two treatments were being compared the two-tailed unpaired Student's *t* test was used; when more than two groups were being compared one-way or two-way ANOVA followed by Tukey's HSD test were used. *p* values <0.05 were considered statistically significant. All the presented data and images have been independently repeated at least three times.

## Reporting summary

Further information on research design is available in the Nature Portfolio Reporting Summary linked to this article.

# Data availability

The RNA-seq data used in this study are deposited in the Gene Expression Omnibus (GEO) database repository under the accession numbers GSE147720, GSE164279, and GSE164316. Source data are provided with this paper.

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

## Acknowledgements

We thank Dr. Pingchang Yang (McMaster University, Hamilton, Canada) for sharing the IEC4.1 cells, Dr. Guan Zhu (Texas A&M University, College Station, TX) for providing the rabbit antiserum against C. parvum membrane proteins, and Dr. Dong-Er Zhang (Department of Pathology, University of California San Diego, La Jolla, California) and Dr. Jenny P. Ting (Department of Genetics, Center for Translational Immunology, University of North Carolina at Chapel Hill, Chapel Hill, North Carolina) for their helpful comments on the study. This work was supported by funding from the National Institutes of Health (AI116323, AI136877, AI141325, and AI156370) to X-M.C. The content is solely the responsibility of the authors and does not necessarily represent the official views of the National Center for Research Resources or the National Institutes of Health.

## Author contributions

X.-M.C. conceived, designed, and supervised the study with the help from M.B. on the CSpV1 virology and viral RNA transfection. S.D. performed most of the experiments and data analysis. W.H., A.-Y.G., M.L., Y.W., Z.X., and X.-T.Z. performed some experiments, helped with the experimental design and performance, analyzed the data, and provided analytical ideas. A.S.H. developed and provided the intestinal tissues from the human tissue bank and helped with data analyses. A.N. helped with the bioinformatics analysis. M.J., P.C.S., K.M.D., and J.K.S. provided critical comments and contributed to study design. S.D. and X.-M.C. wrote the manuscript with input from all authors.

## Competing interests

The authors declare no competing interests.
