## [Peer Review File · Nature Communications]

Cryptosporidium uses CSpV1 to activate host type I interferon and attenuate antiparasitic defensesReviewers' comments:

Reviewer #1 (Remarks to the Author):

In this manuscript, Deng et al demonstrate that interferon I/III signaling may be affecting interferon II control of cryptosporidium. They further implicate a dsRNA virus, that the parasite harbors, as the trigger for interferon I/III signaling. Although well written and highly intriguing, there were many discrepancies in the manuscript that detracted from its overall impact.

Major and minor concerns:

Line 53 – The assertion that *Cryptosporidium* is “minimally invasive” should be removed. I’m not sure if the authors are referring to the parasite’s restriction to epithelial cells or that it lives extra-cytoplasmically in the cell, but this wording can confuse the reader into missing the severity of a diarrheal disease that kills children.

Lines 61-63 – ‘Ligates’ is not appropriate for the description of the interaction of IFN γ with its receptor

Lines 73-75 – The wording should be altered to include the possibility (and likelihood) that IFN lambda, type III interferon, is the predominant response triggered during infection. The type I and type III response are difficult to distinguish, and these referenced publications do not attempt to do so.

Figure 1a – I’m not sure this is necessary so I would encourage the authors to consider whether this adds to the manuscript. If decided yes, please show loxP excision in the figure. Were the other organs used at all in this manuscript?

Figure 1C – Why is intestinal damage the same in *Ifnar*^{-/-} and *Ifnar1fl/fl*?

Figure 1b & d – From legend ‘data is represented by three independent experiments’, but there are only 4 data points for each bar? *See general note below*

Figure 1 – Showing 18s and Hsp70 PCR data is a little redundant. If the authors have the slides from the infected mouse intestine, why not have a (blinded) colleague count parasites?

Figure 2a – Why not used the same genes for the in vitro and in vivo data sets?

Figure 2a – The ‘special upregulated genes’ and ‘other downregulated genes’ might be best used in supplementary figures. Further, the wording of ‘special upregulated genes’ and ‘other down-regulated genes’ might be a bit clearer to the reader as ‘misc upregulated genes’ and ‘misc down-regulated genes’.

Figure 2a – It feels as though the RNAseq here is a bit wasted as the authors are only showing cherry picked genes. I would suggest including functional annotation bioinformatics, or some other unbiased approach to look at pathways and go-terms enriched in their data sets. These could, at the very least, be included as supplementary figures.

Figure 2b – Again, it is becoming increasingly clear that epithelial cells primarily produce and respond to IFN lambda, this includes in response to cryptosporidium. Other groups have noted little IFN α made by epithelial cells. Have the authors confirmed the levels of IFN α , IFN β , and IFN γ being produced in their IEC4.1 cells by ELISA (not RT-PCR)? It looks like lambda isn’t measured here.

Figure 3a – While the data that the authors have previously presented strongly suggest that *Cryptosporidium* injects ncRNAs into its host cell, there have been no independent verifications of this phenomenon and there is no described mechanism. Therefore, I would caution on using these ncRNA as controls. Similarly, COWP1 and CpACBP are poor controls as they are expressed by the parasite in later stages on infection (little to no RNA at 24hrs when this assay was performed).

Figure 3 b and c – Why is there no *C. parvum* counterstain used in b and why is there no dsRNA signal from within the parasite in c? This is a very strange result to find no signal from within the parasite where the virus would be replicating. The probes used appear to find the parasite in b (I assume the large mass of signal here is the parasitophorous vacuole), but there is nothing in c.

Figure 4c – Very strange that transfection of cells with control dsRNA is vastly different than transfection with the dsRNA cryptosporidium virus. This would suggest that the sequence of the virus is making a massive difference in the immune response? And infection with the whole parasite results in expression closer to transfection with virus?...suggesting that the immune response to the parasite is solely to the virus? Again, very surprising result.

Figure 5a and b - It feels like a logical experiment would be to test whether type I interferon addition was capable of reversing the IFN γ blunted infection in the IEC cells (and what effect it has on infection)..?

Figure 5d – The GBP family are induced by interferons, so strange to see them in this table.

Lines 214 to 216 – ‘Knockdown of Ddx58, Mavs, or Eif2ak2 in IEC4.1 cells completely blocked *C. parvum* and CSpV1-dsRNA-induced upregulation of type I IFNs and type I IFN stimulated genes’ Complete block is quite an amazing result. Strange it is not in the main body of the manuscript. Also, PKR (Eif2ak2) knockout completely abrogated IFN β - PKR is usually required for MDA5-mediated, not RIG-I-mediated, IFN production and the authors show RIG-I (Ddx58). MDA5 is also not included, whereas RIG-I is generally considered to be more important for short 10-300bp dsRNA and MDA5 for long (>1000bp) dsRNA.... cryptovirus should be recognized by MDA5...? Including MDA5 and TLR3 (a well known dsRNA receptor) Knockdown in this dataset would likely clarify these results.

Lines 220-221 – ‘Intracellular co-localization of J2 staining was evident with Pkr, RIG-I, and Mavs by fluorescent staining’ – there does not appear to be evidence for colocalization in SupFig 4c.

Figure 6a – Indinavir targets a HIV protease that the cryptovirus does not have. What is the proposed mechanism for restriction here?

Figure 6d – This is a very small change for IFN β and I would again suggest the authors look at IFN β levels

Lines 243-244 and 316-318 – Wording describing synergy should be softened here – synergy literally means that the combined effect is greater than the sum of their separate effects. The inhibition seen is equivalent to the sum of the effects of paromomycin and ribavirin alone.

Lines 283-284 – Semantic note, but without knowing the effect. Of the virus on the parasite how can we make the statement that the virus is a symbiont?

Lines 288-289 – ‘LRV1 within the *Leishmania* parasite is transferred into host cells and exacerbates the pathogenesis of human leishmaniasis’ I’m unable to find the data that shows LRV1 is transferred into the cytosol of the host. LRV1 is detected

Extended data Fig 4a – Why is there an siControl for this experiment?

Extended data Fig b – I’m unclear why NR_045064 is a positive control

*General note – Frequently, the authors are referring to ‘three independent experiments’, but I believe they mean ‘three biological replicates’ unless each experiment had one biological replicate. The spread of the data also suggests biological replicates instead of independent. This should be cleared up prior to any consideration of publication.

Reviewer #2 (Remarks to the Author):

The manuscript by Deng et al describes that the delivery of Cryptosporidium virus (CSpV1) double stranded RNA into host cell upon infection activates type I IFN signaling and attenuates IFN-gamma (IFN-g) mediated epithelial anti-parasitic response, thus allowing the parasite to establish infection. Conditional KO mice that are deficient in type I IFN receptor (IFNar1^{-/-}) showed resistance to infection. The authors performed transcriptomics on ileal epithelium from infected and uninfected IFNar1^{-/-} cKO mice and cultured IEC4.1 cells deficient in IFNar1 and found downregulation of the type I/III IFN-stimulated genes. It is not clear why the authors picked this particular cell line and not organoids for all their studies. Although findings presented in this manuscript are interesting, previous studies on transcriptomics analysis of *C. parvum* infected organoids/miniguts have demonstrated signatures of type I IFN signaling responses (Heo et al 2018 Nature Microbiology; Nikolaev et al 2020 Nature). Moreover, the authors did not discuss these previous transcriptomics studies and how their results compare to these datasets. A major caveat of the study is that the cross talk between type I and type III (IFN- λ) signaling or the role of type III IFN in control of *C. parvum* infection could not be established.

My other major concerns are as follows:

- No quantification of Cryptosporidium burden in H&E stained intestinal epithelial is included and only representative images are provided (Fig.1C)
- RNAseq analysis of IFNar1^{-/-} mice infected with *C. parvum* (48h) replicates are not consistent and seem similar to uninfected mice replicates (Fig,2). Volcano plots could have been provided to visualize top up/down regulated genes. There is no description of how the RNAseq data was analyzed and details on number of reads, coverage, filtering, FDR, log FC, statistical analysis etc are missing.
- In Fig. 4F, there is no measurement of parasite growth.
- Fig. 6 does not add much to the results. It is not discussed why there was no effect with the antiviral indinavir? Also, since infection was done for 2 hours followed by drug treatment is it possible that the infection was not fully established and drug treatment was initiated? The immunofluorescent staining is not very clear, and Cryptosporidium can be hardly seen making it difficult to visualize parasite burden.

Reviewer #3 (Remarks to the Author):

This outstanding research reports that *C. parvum* delivers to its host cell a double-stranded virus named Cryptosporidium parvum virus (CSpV1) which induces a strong type I interferon response. First, this manuscript demonstrates that mice lacking type I IFN receptor (villin.ifnar1^{-/-}) in the intestinal cells are resistant to *C. parvum* infection at 48 and 72 hpi. However, a significant difference in in vitro infection of IEC4 Ifnar1^{-/-} cells is not observed.

Second, gene expression analysis between *C. parvum* infected and uninfected villin.ifnar1^{-/-} mice, control mice, IEC4 Ifnar1^{-/-} cells and control cells, shows differences at the type I/III IFN signaling. Including, IFN-gamma induced genes, cytokines-chemokines, cytoskeleton, cell proliferation and apoptosis.

Third, the authors demonstrate the delivery of CSpV-1 double stranded RNAs into the cytoplasm of infected IEC4 host cells after infection with *C. parvum* by different techniques such as RT-PCR, in situ hybridization, immunofluorescence, dot blotting and pull-down assays.

Moreover, IEC4 cells or human intestinal organoids transfected with CSpV1s induce the expression type I IFN and type I/III IFN stimulated genes similar to a *C. parvum* infection. The knocking-down of CSpV1-dsRNAs in IEC4 cells reduces type I IFN gene expression. Also, this research confirms that CSpV1 ds-RNAs induces type I IFN gene expression in infected cells by activating the PKR/RIG-1/MAVS signal pathway.

Finally, the research evaluates that the anti-viral drug Indinavir, in HCT8 cells, and the anti-viral drug Ribavirin, in mice, are effective in combination with the anti-parasitic drug paromomycin reducing parasite burden.

In conclusion, all these findings are significant to the field of parasitology because they provide advancement in our knowledge of the parasite for the development of better anti-cryptosporidium treatments. This work supports the conclusions and claims of the authors. The methodology was implemented rigorously. There are statements in the result section that need to be improved as well as some experiments that need more details. This manuscript requires the following editions:

1. The Reference 7 has switched order of author's name and last name.
2. Line 74. This reference (PMID 34928716) might strengthen your claims because it explains that Interferon-type I response reduces shedding of oocysts but does not affect parasite burden in immunocompromised mice.
3. The Legend of extended data Fig 1 said PNCA instead of PCNA (proliferating cell nuclear antigen). The same typo is observed in Figure 1c, in Line 93 and in line 741 of the manuscript.
4. Why is the X-axis title of the RT-PCR graphs labeled "RNA level" instead of "gene expression"? Also, some figures are labeled "ratio to control" and others "Fold to control", but in the legend or methods is not explained how this was calculated. For example, in the Extended data Fig 2a and 2c. Additionally, it would be easier for the reader to observe the two bars that are being compared by statistical analysis and not only a *. Also, please, indicate when it is not significant.
5. The image of the western blot in Extended data Fig 2b is not clear. Please, consider adding a higher resolution image.
6. Fig 1b needs to be cited in line 113 to better understand the sentence.
7. It is not clear in the methods how the mice samples for RNA sequencing were collected. How were the ileal epithelial samples collected? Were the samples concentrated or filtered? The methodology affects the results.
8. Line 126-128. The statement is not true. The heatmap in Figure 2a, shows low abundance of some genes related to type I/III IFN stimulated genes in the ilea of infected ifnar1^{-/-} mice at 48 hpi but only a mouse at 72 hpi. Also, it is not clear how many biological replicates or mice were used for this experiment, does a column in the heatmap represent a mouse?
9. Line 128-130. I do not agree with this statement because the heatmap in Fig 2a, does not show lower expression of IFN γ .
10. Line 130-131. I do not agree with this statement because the heatmap in Fig 2a, does not show lower expression of IFN γ -stimulated genes in infected ifnar1^{-/-} mice at 72 hpi.
11. Line 131-134. I do not agree with this statement because the heatmap in Fig 2a does not show higher expression of "special upregulated genes" in infected ifnar1^{-/-} mice. I can observe a lot of variability. This trend is not clear in the heatmap. If you want to emphasize these results, I suggest presenting in other kind of figure where the effect can be clearly observed.
12. The colors of Fig 2b are not easily distinguished.
13. It is not clear what time point was used in Fig 2b. Does it show the infected neonate samples at 48 hpi or 72 hpi? Does each dot in the bars represent a technical or biological replicate?
14. What is the control in Fig 3a?
15. In line 164 and 165, it is not clear why this is a control of parasite lysis.
16. Fig 4c is missing an explanation of analyzed genes and the criteria for dendrogram association.
17. The legend of the figure 4f and 4g requires more details.

POINT-BY-POINT RESPONSES TO THE REVIEWERS' COMMENTS

We thank all the reviewers for your positive comments on our manuscript. We appreciate your thoughtful review and have attempted to satisfactorily respond to your concerns and suggestions. The major concern raised by all reviewers about the role (or cross-link) for type I and type III IFN signaling in control of *Cryptosporidium* infection. To address this concern, we have performed additional experiments on the induction of type III IFNs in the intestinal epithelium following infection and have extensively revised our manuscript to address this important point. With the additional data in the revised manuscript, our findings demonstrate that all three types of IFN signaling are activated in the neonatal intestinal epithelium following infection. Activation of type I IFN signaling in intestinal epithelial cells contributes significantly to the transcriptomic alterations in the infected intestinal epithelium, including transactivation of the type I/III-stimulated genes, at least during the early infection stage in neonatal mice. Therefore, each type of IFNs displays distinct roles in intestinal immunity to *Cryptosporidium* infection. Whereas type II and III IFNs show a profound protective role in control the infection, type I IFN signaling in the intestinal epithelial cells is detrimental to intestinal anti-*C. parvum* defense. Coupled with our observation on the activation of type I IFN response in intestinal epithelium through delivery of CSpV1-dsRNAs, our data indicate a detrimental effect of type I IFN signaling in intestinal epithelial cells on intestinal anti-*C. parvum* defense and reveal a new strategy of immune evasion by the parasite.

Responses to comments from Reviewer #1

In this manuscript, Deng et al demonstrate that interferon I/III signaling may be affecting interferon II control of *Cryptosporidium*. They further implicate a dsRNA virus, that the parasite harbors, as the trigger for interferon I/III signaling. Although well written and highly intriguing, there were many discrepancies in the manuscript that detracted from its overall impact. ---- Thank you so much for your positive comments and thoughtful insights and suggestions about our manuscript. We highly valued your suggestions and as listed below, have performed additional experiments and revised the manuscript to address all your concerns.

Major and minor comments:

- 1) The assertion that *Cryptosporidium* is “minimally invasive” should be removed. I’m not sure if the authors are referring to the parasite’s restriction to epithelial cells or that it lives extra-cytoplasmically in the cell, but this wording can confuse the reader into missing the severity of a diarrheal disease that kills children. ---- We have removed this sentence from the text to avoid confusion to the readers.
- 2) ‘Ligates’ is not appropriate for the description of the interaction of IFN γ with its receptor. ---- We have modified this sentence for the description of the interaction of IFN γ with its receptor in the revised manuscript (line 66).
- 3) The wording should be altered to include the possibility (and likelihood) that IFN lambda, type III interferon, is the predominant response triggered during infection. The type I and type III response are difficult to distinguish, and these referenced publications do not attempt to do so. ---- That’s a great point and we totally agree with you. We have added several additional references and included discussion about the role for type III interferons in host defense during *Cryptosporidium* infection (lines 67-82).
- 4) I’m not sure this is necessary so I would encourage the authors to consider whether this adds to the manuscript. If decided yes, please show loxP excision in the figure. Were the

- other organs used at all in this manuscript? ---- We have removed this part to the Extended Data Fig. 1a. None of the other organs were used for this study.
- 5) **Why is intestinal damage the same in *lfnar*^{-/-} and *lfnar1fl/fl*?** ---- RNA-Seq analysis revealed comparable expression levels for many genes in the intestinal epithelium from infected *Villin.lfnar1^{-/-}* and *lfnar1^{fl/fl}* mice, including many inflammatory cytokine and chemokine genes and genes that code proteins for cytoskeleton and cell proliferation and apoptosis. This may account for the similar morphological changes observed in the infected intestine from *Villin.lfnar1^{-/-}* and *lfnar1^{fl/fl}* mice. We have included this discussion in the revised manuscript (lines 169-173).
 - 6) **From legend ‘data is represented by three independent experiments’, but there are only 4 data points for each bar? *See general note below*.** --- Sorry for our ambiguous description about this. It is true that some data are from three or four independent experiments whereas others are from three biological replicates. We have clarified this in all the legends for each experiment for the entire manuscript.
 - 7) **Figure 1 – Showing 18s and Hsp70 PCR data is a little redundant. If the authors have the slides from the infected mouse intestine, why not have a (blinded) colleague count parasites?** ---- We thank you for your suggestion. We have moved the cpHsp70 PCR data to the Extended Data Fig. 1d and added the parasite counting data in the new Fig. 1b.
 - 8) **Figure 2a – Why not used the same genes for the in vitro and in vivo data sets?** ---- We have completed reorganized the Fig. 2 to include more bioinformatics analyses of the RNA-seq data sets. Accordingly, comparison between gene expression profiles in the in vitro and in vivo data sets were presented in the revised Fig. 2 and expression levels of representative genes were added in the Extended Data Fig. 3.
 - 9) **Figure 2a – The ‘special upregulated genes’ and ‘other downregulated genes’ might be best used in supplementary figures. Further, the wording of ‘special upregulated genes’ and ‘other down-regulated genes’ might be a bit clearer to the reader as ‘misc upregulated genes’ and ‘misc down-regulated genes’.** ---- That’s a great idea. We have moved the “Cytokines and chemokines”, “Cytoskeleton”, “Cell proliferation and apoptosis”, and “other down-regulated genes” from Fig. 2a and presented them in the Extended Data Table 1 and 2. We have also used the terms “misc upregulated genes” and “misc down-regulated genes” in the revised Fig. 2e and Extended Data Fig. 3c.
 - 10) **Figure 2a – It feels as though the RNA-seq here is a bit wasted as the authors are only showing cherry picked genes. I would suggest including functional annotation bioinformatics, or some other unbiased approach to look at pathways and go-terms enriched in their data sets. These could, at the very least, be included as supplementary figures.** ---- We thank you for your suggestions. We have added more bioinformatics on the RNA-seq data in Fig. 2 and the new Extended Data Fig. 3.
 - 11) **Figure 2b – Again, it is becoming increasingly clear that epithelial cells primarily produce and respond to IFN lambda, this includes in response to *Cryptosporidium*. Other groups have noted little IFN α made by epithelial cells. Have the authors confirmed the levels of IFN α , IFN β , and IFN λ being produced in their IEC4.1 cells by ELISA (not RT-PCR)? It looks like lambda isn’t measured here.** ---- That’s an important point. Intestinal epithelial cells primarily produce and respond to IFN lambda, this includes in response to *Cryptosporidium* as demonstrated by two recent reports (Gibson AR, et al. PLoS Pathogens, 2022 and Ferguson SH et al., Cell Mol Gastroenterol Hepatol. 2019). Per your suggestion, we have performed additional experiments to measure the levels of IFN- α , IFN- β , and IFN- λ being produced in in vitro infected IEC4.1 cells and in vivo infected intestinal epithelium either by RT-qPCR or by ELISA. Taken together, our data demonstrate that all three types of IFN signaling are activated in the intestinal epithelium following

infection in vivo. Besides type II and III IFNs, a significant amount of type I IFNs is also produced in the intestinal epithelium and activation of type I IFN signaling significantly contributes to the upregulated transcription of the type I/III-stimulated genes in the intestinal epithelium following *Cryptosporidium* infection in mice, particularly during the early infection stage. Overall, as described above, our data demonstrate that type I IFN signaling displays a distinct role in intestinal immunity to *Cryptosporidium* infection. Whereas type II and III IFNs show a profound protective role in control the infection, type I IFN signaling in the intestinal epithelial cells displays a pro-parasitic effect. We have included the new data and extensively revised the whole manuscript to address this important point (new Fig. 2a and b and lines 131-189).

- 12) **Figure 3a – While the data that the authors have previously presented strongly suggest that *Cryptosporidium* injects ncRNAs into its host cell, there have been no independent verifications of this phenomenon and there is no described mechanism. Therefore, I would caution on using these ncRNA as controls. Similarly, COWP1 and CpACBPB are poor controls as they are expressed by the parasite in later stages on infection (little to no RNA at 24hrs when this assay was performed). ---- That's very thoughtful insights. We have deleted the data on these parasite ncRNAs and added new data on the RNA levels of these genes expressed in early stages on infection as previously demonstrated (mBio 2020;11:e00052-20 and Nat Microbiol 2019;4:2226-2236). Our data continue to support the delivery of CSpV1-dsRNAs into infected host cells. We have included the new data in the revised manuscript (new Fig. 3a).**
- 13) **Figure 3 – Why is there no *C. parvum* counterstain used in b and why is there no dsRNA signal from within the parasite in c? This is a very strange result to find no signal from within the parasite where the virus would be replicating. The probes used appear to find the parasite in b (I assume the large mass of signal here is the parasitophorous vacuole), but there is nothing in c. ---- We have performed additional staining to use various methodologies to increase the membrane permeability and collected new images showing the signaling of CSpV1-dsRNAs in the parasites as well in the new Fig. 3c.**
- 14) **Figure 4c – Very strange that transfection of cells with control dsRNA is vastly different than transfection with the dsRNA cryptosporidium virus. This would suggest that the sequence of the virus is making a massive difference in the immune response? And infection with the whole parasite results in expression closer to transfection with virus?...suggesting that the immune response to the parasite is solely to the virus? Again, very surprising result. ---- Sorry for the confusion regarding the control dsRNA that we used for the experiment. The control dsRNA we used for control was a non-specific scrambled double-stranded siRNA, not a long non-specific dsRNA with the same length as the CSpV1-dsRNAs. Therefore, it is reasonable to see the control showing vastly different from the transfection with the CSpV1-dsRNAs. In addition, significant difference was observed in the expression profiles in cells following *Cryptosporidium* infection vs cells transfected with the CSpV1-dsRNAs, whereas the type I IFN response was comparable. We have included a more detailed description of the control dsRNA and also discussion about the similarity and difference between the gene expression profiles of the two groups (infected cells vs cells transfected with the CSpV1-dsRNAs) in the revised manuscript (new Fig. 4c, d, e and lines 231-241).**
- 15) **Figure 5a and b - It feels like a logical experiment would be to test whether type I interferon addition was capable of reversing the IFN γ blunted infection in the IEC cells (and what effect it has on infection)? ---- That's a great point. Previous studies already demonstrated that pre-treatment of epithelial cells with IFN- α before exposure to *C. parvum* could modestly inhibit parasite invasion of host cells (J Infect Dis. 2009;200:1548-55). This will make the resultant data very difficult to be interpreted if we add the type I IFNs to the cell cultures before infection or during the very early infection stage. Therefore, instead, we performed an alternative experiment and pre-treated cells with IFN- α and then measured their response to IFN- γ stimulation. Decreased**

expression levels of several selected IFN- γ -stimulated genes, including *Igtp*, *Irf8*, and *Ifi47*, were observed in our previous experiments in IEC4.1 cells first exposed to *C. parvum* for 24h followed by IFN- γ stimulation for additional 2h. Similarly, suppression of expression of *Igtp* and *Irf8* was also observed in IEC4.1 cells first treated with IFN- α for 8h followed by IFN- γ stimulation for additional 2h. Thus, our new data continue to support that type I IFN signaling may have an inhibitory effect in intestinal epithelial cells on their response to subsequent IFN- β stimulation. We have included the new data in the revised manuscript (Extended Data Figure 5 and lines 286-294).

- 16) **Figure 5d – The GBP family are induced by interferons, so strange to see them in this table.** -
--- Yes, the GBP family are induced by interferons and we previously listed some of them as the positive control genes in the table. To avoid confusion, we have deleted them from this table (Fig. 5d).
- 17) **Lines 214 to 216 – ‘Knockdown of Ddx58, Mavs, or Eif2ak2 in IEC4.1 cells completely blocked *C. parvum* and CSpV1-dsRNA-induced upregulation of type I IFNs and type I IFN stimulated genes’ Complete block is quite an amazing result. Strange it is not in the main body of the manuscript. Also, PKR (Eif2ak2) knockout completely abrogated IFN β - PKR is usually required for MDA5-mediated, not RIG-I-mediated, IFN production and the authors show RIG-I (Ddx58). MDA5 is also not included, whereas RIG-I is generally considered to be more important for short 10-300bp dsRNA and MDA5 for long (>1000bp) dsRNA..... cryptovirus should be recognized by MDA5...? Including MDA5 and TLR3 (a well known dsRNA receptor) Knockdown in this dataset would likely clarify these results. ---- Our data indicate that knockdown of Ddx58, Mavs, or Eif2ak2, but not Mda5, in IEC4.1 cells completely blocked *C. parvum* and CSpV1-dsRNA-induced upregulation of type I IFNs and type I IFN stimulated genes. Per your suggestion, we have performed experiments testing the impact of TLR3 knockdown on *C. parvum*- and CSpV1-dsRNA-induced upregulation of type I IFNs and type I IFN stimulated genes. We took the siRNA approach to knockdown TLR3 and no significant inhibition of *C. parvum*- and CSpV1-dsRNA-induced upregulation of *Ifnb1* (a type I IFN stimulated gene) was detected. Interestingly, *Cryptosporidium* infection triggers strong type III response in intestinal epithelium through TLR3-dependent recognition (PLoS Pathog, 2012;8:e1002670). Therefore, distinct mechanisms are involved in the activation of type I and III responses in host cells following *Cryptosporidium* infection. We have included this point in our discussion (lines 256-259 and 401-405).**
- 18) **‘Intracellular co-localization of J2 staining was evident with Pkr, Rig-I, and Mavs by fluorescent staining’ – there does not appear to be evidence for colocalization in SupFig 4c.** -
--- We agree that fluorescent staining images do not directly support colocalization and thus, have deleted this part from the revised manuscript.
- 19) **Figure 6a – Indinavir targets a HIV protease that the cryptovirus does not have. What is the proposed mechanism for restriction here?** ---- Thank you for pointing out this point. Although Indinavir targets a HIV protease that the cryptovirus does not have, resultants from several previous studies also support that it can reduce *Cryptosporidium parvum* infection in both in vitro and in vivo models but underlying mechanisms remain unclear (J Antimicrob Chemother. 2003;52:359-64). Interestingly, the only FDA-approved anti-*C. parvum* drug nitazoxanide also displays anti-viral activity in chronic hepatitis B and C (Antiviral Res. 2008;77:56-63 and Cell Mol Gastroenterol Hepatol. 2019;7:297-312). We have included discussion regarding this in the revised manuscript (lines 418-426).
- 20) **Figure 6d – This is a very small change for IFN β and I would again suggest the authors look at IFN λ levels.** ---- Per your suggestion, we performed additional experiments to measure the IFN λ levels. We have included the new data and also discussion on this important point in the revised manuscript (new Fig. 7d and lines 306-309).

- 21) Lines 243-244 and 316-318 – Wording describing synergy should be softened here – synergy literally means that the combined effect is greater than the sum of their sperate effects. The inhibition seen is equivalent to the sum of the effects of paromomycin and ribavirin alone. -- We have modified the text (lines 296-317).
- 22) Lines 283-284 – Semantic note, but without knowing the effect. Of the virus on the parasite how can we make the statement that the virus is a symbiont? ---- We have revised the description text accordingly throughout the entire manuscript.
- 23) Lines 288-289 – ‘LRV1 within the Leishmania parasite is transferred into host cells and exacerbates the pathogenesis of human leishmaniasis’ I’m unable to find the data that shows LRV1 is transferred into the cytosol of the host. LRV1 is detected. ---- We have corrected this mistake (line 408-412).
- 24) Extended data Fig 4a – Why is there an siControl for this experiment? ---- We have included data on *Tlr3* siRNA knockdown to this figure and thus, wanted to keep this control in the revised figure (now Fig. 5b, c).
- 25) Extended data Fig b – I’m unclear why NR_045064 is a positive control. ---- We have deleted this from the revised manuscript.
- 26) *General note – Frequently, the authors are referring to ‘three independent experiments’, but I believe they mean ‘three biological replicates’ unless each experiment had one biological replicate. The spread of the data also suggests biological replicates instead of independent. This should be cleared up prior to any consideration of publication. ---- As described above in our response in concern #6, we have clarified this for all the experiments in the manuscript.

Responses to comments from Reviewer #2

The manuscript by Deng et al describes that the delivery of *Cryptosporidium* virus (CSpV1) double stranded RNA into host cell upon infection activates type I IFN signaling and attenuates IFN-gamma (IFN- γ) mediated epithelial anti-parasitic response, thus allowing the parasite to establish infection. Conditional KO mice that are deficient in type I IFN receptor (IFNar1^{-/-}) showed resistance to infection. The authors performed transcriptomics on ileal epithelium from infected and uninfected IFNar1^{-/-} cKO mice and cultured IEC4.1 cells deficient in IFNar1 and found downregulation of the type I/III IFN-stimulated genes. It is not clear why the authors picked this particular cell line and not organoids for all their studies. Although findings presented in this manuscript are interesting, previous studies on transcriptomics analysis of *C. parvum* infected organoids/miniguts have demonstrated signatures of type I IFN signaling responses (Heo et al 2018 Nature Microbiology; Nikolaev et al, 2020 Nature). Moreover, the authors did not discuss these previous transcriptomics studies and how their results compare to these datasets. A major caveat of the study is that the cross talk between type I and type III (IFN-lambda) signaling or the role of type III IFN in control of *C. parvum* infection could not be established. ---- We thank you for your positive comments and thoughtful insights and suggestions about our manuscript. Signatures of type I IFN responses in response to *Cryptosporidium* infection have been demonstrated in previous studies (J Infect Dis. 2009;200:1548-55). Our findings from this study provide additional information highlighting the potential mitigate effects of type I IFN response on host anti-*Cryptosporidium* defense in the neonatal intestine. Due to our focus on neonatal intestine, we used an in vivo infection model of neonatal mice, ex vivo model of organoids/2D monolayers isolated from neonates, and in vitro infection model employing the IEC4.1 cells (an intestinal epithelial cell line derived from neonatal mice) for our analysis. We have added this rationale, as well as comparison of gene expression profiles with previous transcriptomics studies, in the revised manuscript (lines 118-119 and 159). Moreover, to clarify whether

type III IFNs are also involved in control of infection in neonatal intestine, we have performed additional experiments to measure the expression of type III IFNs in our models. Taken together, our data demonstrate that all three types of IFN signaling are activated in the intestinal epithelium following infection *in vivo*. Besides type II and III IFNs, a significant amount of type I IFNs is also produced in the intestinal epithelium and activation of type I IFN signaling significantly contributes to the upregulated transcription of the type I/III-stimulated genes in the intestinal epithelium following *Cryptosporidium* infection, particularly during the early infection stage. In addition, each type of IFNs displays distinct role in intestinal immunity to *Cryptosporidium* infection. Whereas type II and III IFNs show a profound protective role in control the infection, type I IFN signaling in the intestinal epithelial cells displays a pro-parasitic effect. We have included the new data and discussion on this important point in the revised manuscript (new Fig. 2a and b and lines 131-189 and 320-349).

Other major concerns:

- 1) **No quantification of *Cryptosporidium* burden in H&E stained intestinal epithelial is included and only representative images are provided (Fig.1C)** ---- We thank you for your suggestion and have added the parasite counts from the infected mouse intestine from a blinded colleague (new Fig. 1b).
- 2) **RNAseq analysis of IFN α 1-/- mice infected with *C. parvum* (48h) replicates are not consistent and seem similar to uninfected mice replicates (Fig. 2). Volcano plots could have been provided to visualize top up/down regulated genes. There is no description of how the RNAseq data was analyzed and details on number of reads, coverage, filtering, FDR, log FC, statistical analysis etc are missing.** ---- That's a great idea and thank you. We have included more bioinformatics on the RNA-seq data in Fig. 2 and in the new Extended Data Fig. 3, including functional annotation, GSEA signaling pathway analyses. We have also added detailed description of how the RNA-Seq data was analyzed in the revised manuscript (lines 563-571).
- 3) **In Fig. 4F, there is no measurement of parasite growth.** ---- That's a great point and we have measured the parasite counts in the cells and no significant difference was detected at this timepoint (early attachment/invasion phase) following siRNA-POOLS treatment, suggesting that the decrease of CSpV1-dsRNA levels was not due to a decreased parasite infection. We have included the new data and discussion on this important point in the revised manuscript (new Fig. 4g and lines 246-247).
- 4) **Fig. 6 does not add much to the results. It is not discussed why there was no effect with the antiviral indinavir? Also, since infection was done for 2 hours followed by drug treatment is it possible that the infection was not fully established and drug treatment was initiated? The immunofluorescent staining is not very clear, and *Cryptosporidium* can be hardly seen making it difficult to visualize parasite burden.** ---- We have performed additional experiments testing the effect of antiviral reagents using the 2D human intestinal epithelial monolayers. Resultant data continue to support that antiviral reagents can promote the inhibition of antiparasitic drugs on *C. parvum* infection of intestinal epithelium. Therefore, we wanted to keep this figure with the new data from the 2D human monolayers and also included discussion addressing your concern in the revised manuscript (lines 299-301 and 494-511). In addition, we have added insets of higher magnifications to the boxed regions of these images to better show the parasite staining (Fig. 7g).

Responses to comments from Reviewer #3

This outstanding research reports that *C. parvum* delivers to its host cell a double-stranded virus named *Cryptosporidium parvum* virus (CSpV1) which induces a strong type I interferon response.

First, ---. Second, ---. Third, ---. Finally, ---. In conclusion, all these findings are significant to the field of parasitology because they provide advancement in our knowledge of the parasite for the development of better anti-*Cryptosporidium* treatments. This work supports the conclusions and claims of the authors. The methodology was implemented rigorously. There are statements in the result section that need to be improved as well as some experiments that need more details. --- Thank you so much for your positive comments and thoughtful insights and suggestions about our manuscript. We highly valued your suggestions and as listed below, have performed additional experiments and modified the manuscript text accordingly to address all your concerns.

This manuscript requires the following editions:

- 1) **The Reference 7 has switched order of author's name and last name. --- We have corrected this mistake.**
- 2) **This reference (PMID 34928716) might strengthen your claims because it explains that Interferon-type I response reduces shedding of oocysts but does not affect parasite burden in immunocompromised mice. --- That's an important point and thank you. We have added this reference and included discussion about this point in the revised manuscript (lines 77-79).**
- 3) **The Legend of extended data Fig 1 said PNCA instead of PCNA (proliferating cell nuclear antigen). The same typo is observed in Figure 1c, in Line 93 and in line 741 of the manuscript. --- We have corrected these mistakes.**
- 4) **Why is the X-axis title of the RT-PCR graphs labeled "RNA level" instead of "gene expression"? Also, some figures are labeled "ratio to control" and others "Fold to control", but in the legend or methods is not explained how this was calculated. For example, in the Extended data Fig 2a and 2c. Additionally, it would be easier for the reader to observe the two bars that are being compared by statistical analysis and not only a *. Also, please, indicate when it is not significant. --- We have revised the labelling in these figures to be consistent and added explanation about the methods of calculation in the text. In addition, we have also added the statistic significant marks for all the figures to be easier for the readers to observe.**
- 5) **The image of the western blot in Extended data Fig 2b is not clear. Please, consider adding a higher resolution image. --- We have replaced with a higher resolution image in Extended data Fig. 2b.**
- 6) **Fig 1b needs to be cited in line 113 to better understand the sentence. We have cited the Fig. 1b data to this sentence to make it better to understand (line 125).**
- 7) **It is not clear in the methods how the mice samples for RNA sequencing were collected. How were the ileal epithelial samples collected? Were the samples concentrated or filtered? The methodology affects the results. Thank you for your suggestions. We have added more bioinformatics on the RNA-seq data in Fig. 2 and the new Extended Data Fig. 3, including functional annotation, GSEA signaling pathway analyses. We have also added detailed description of how the RNA-Seq data was analyzed in the revised manuscript (lines 563-571).**
- 8) **Line 126-128. The statement is not true. The heatmap in Figure 2a, shows low abundance of some genes related to type I/III IFN stimulated genes in the ilea of infected ifnar1-/- mice at 48 hpi but only a mouse at 72 hpi. Also, it is not clear how many biological replicates or mice were used for this experiment, does a column in the heatmap represent a mouse. --- We thank you for your thoughtful insights. We have performed additional bioinformatics analyses on the gene expression profiles, including more detailed GSEA and Volcano analysis, showing more clearly the suppression of type I/III IFN-stimulated genes in the infected Ifnar1-/- mice, particularly at 48h p.i. (Fig. 2c, d, e, f). We have also added information about how many mice were used for these experiments in the figure legends.**

- 9) **Line 128-130. I do not agree with this statement because the heatmap in Fig 2a, does not show lower expression of IFN γ .** ---- Thank you for your thoughtful insights. We have revised the statement and related description of the data in the revised Figures (Fig. 2d, e, f) and manuscript text (lines 163-165).
- 10) **Line 130-131. I do not agree with this statement because the heatmap in Fig 2a, does not show lower expression of IFN γ -stimulated genes in infected *ifnar1*^{-/-} mice at 72 hpi.** ---- We agree and thank you for your thoughtful insights. We have revised the statement and related description of the data in the revised Figures (Fig. 2d, e, f) and manuscript text (lines 163-165).
- 11) **Line 131-134. I do not agree with this statement because the heatmap in Fig 2a does not show higher expression of “special upregulated genes” in infected *ifnar1*^{-/-} mice. I can observe a lot of variability. This trend is not clear in the heatmap. If you want to emphasize these results, I suggest presenting in other kind of figure where the effect can be clearly observed.** ---- Per your suggestion, we have revised the Fig. 2 with more detailed bioinformatics analyses of the data with a focus on the higher expression of “misc./special upregulated genes” in infected *Ifnar1*^{-/-} mice at 48h post-infection in the revised Figures (Fig. 2e and f) and manuscript text (lines 150-165). Many thanks for your great suggestions.
- 12) **The colors of Fig 2b are not easily distinguished.** ---- We have changed the color patterns to make them easier to be distinguished.
- 13) **It is not clear what time point was used in Fig 2b. Does it show the infected neonate samples at 48 hpi or 72 hpi? Does each dot in the bars represent a technical or biological replicate? --** -- We have added detailed information about these points in the figure legends in the revised manuscript.
- 14) **What is the control in Fig 3a?** --- We have included additional information about the control in the revised manuscript (lines 202-205) and revised legend for Fig. 3a.
- 15) **In line 164 and 165, it is not clear why this is a control of parasite lysis.** --- As described above in our response to concern #12 for the reviewer 1, we have deleted the data about parasite ncRNAs from the manuscript.
- 16) **Fig 4c is missing an explanation of analyzed genes and the criteria for dendrogram association.** ---- We have added detailed information and explanation about this in the revised legend for Fig. 4c.
- 17) **The legend of the figure 4f and 4g requires more details.** ---- We have added more information and explanation about the legend in the revised manuscript.

REVIEWER COMMENTS

Reviewer #1 (Remarks to the Author):

For the most part, the authors have satisfactorily addressed the critique of Reviewer 1. However, they have not directly answered question 5 about why intestinal damage is the same in *Ifnar*^{-/-} and *Ifnar*^{fl/fl} mice? This is an important question that should be answered.

Reviewer #2 (Remarks to the Author):

The authors have made revisions to the manuscript in light of reviewers' comments. Their results corroborates with previously published data showing upregulation of type I IFNs, IFN- γ -mediated signaling and cytokine-mediated signaling pathways. As cited by the authors (and reported by Gibson et al 2022 PLoS Pathogens), *Cryptosporidium* oocyst shedding was significantly reduced in the *Ifnar*^{1-/-} mice compared to wild type. Although they have cited previous studies, the authors did not fully discuss the novelty of their findings and impact of their work.

The authors have not fully highlighted and discussed the novelty or impact of their work. The role of interferons (type I/III and IFN- γ in *Cryptosporidium* infection has been previously reported. Most of their work replicate previously reported findings, with only an incremental increase in new knowledge.

There are discrepancies in the data analysis and interpretation. These include:

There is a discrepancy between the replicates shown in the RNAseq data heatmaps in Figure 4 (Fig. 4C and 4E)

Details on RNAseq analysis is still lacking. Quality controls and cut-offs for assessing poor quality reads are not discussed.

Poor quality images of immunofluorescence staining of intestinal epithelium. It is hard to see parasites in the representative image. Not clear how the authors managed to count parasites in these images.

The exact mechanism of action of nitazoxanide is not known, therefore hinging on its potential antiviral activity may be an over-interpretation of data

Reviewer #3 (Remarks to the Author):

Deng et al, reports that *C. parvum* induce type I IFN response by delivery of CSpV1- dsRNAs to its host. Also, author analyzed the activation of pathway PKR/ RIG/MAV/STING by CSpV1- dsRNAs. Finally, the authors study the effects of anti-viral reagents and anti-parasite drugs on *C. parvum* infection of intestinal epithelium. Deng et al have addressed all previous suggestions and the manuscript have improved considerably. there are seven aspects that still need to be improved:

1. Fig 2b says *C. parvum* (48 h) when the legend and the manuscript say 24 h. Please correct and clarify. Fig 2b Legend say two-tail, it should be changed to two-tailed. Fig 2e shows the data at 24 hpi but the manuscript (line 161) and the legend of the figure says 48 hpi, please correct.
2. Please re-write lines 146-153. It is not clear what you are comparing in Fig 2c, only uninfected mice?
3. Line 171. Please, cite the correct figure.
4. Line 193. Please, improve sentence.
5. It will be great to have an illustration like reference 37 or propose a model what is happening in *C. parvum* and their CSpV1-dsRNAs.

6. In line 339, reference 17, please add that IFN type I response reduced oocyst shedding and symptoms but not intestinal parasite burden in IFN γ KO mice.
7. In line 348. Please, explain why IFN type I display pro-parasitic effects.
8. Normally, the quality of reads is determined by (FASQC), a threshold is selected and only reads that meet the thresholds are used for further analysis. Also, the data normally is trimmed to remove low-quality reads (Trimmomatic). Please supply more details on the RNAseq- analysis.

POINT-BY-POINT RESPONSES TO THE REVIEWERS' COMMENTS

Responses to comments from Reviewer #1

For the most part, the authors have satisfactorily addressed the critique of Reviewer 1. However, they have not directly answered question 5 about why intestinal damage is the same in *Ifnar*^{-/-} and *Ifnar*^{fl/fl} mice? This is an important question that should be answered ----- Thank you for your positive comments about our revisions. We agree with you that it is important to clarify why intestinal damage is similar in the infected *Villin.Ifnar*^{1-/-} and *Ifnar*^{1^{fl/fl}} mice, given a higher infection burden (parasite replication) was observed in the *Ifnar*^{1^{fl/fl}} mice. Infection of the intestinal epithelium with *Cryptosporidium* can result in intestinal morphological changes and damage such as blunting of intestinal villi and crypt hyperplasia (Argenzio RA, et al. 1990, Gastroenterol). Inflammation and apoptotic cell death are the main effects for *Cryptosporidium*-induced intestinal damage during acute infection. Previous studies demonstrated both pro-apoptotic and anti-apoptotic effects of *Cryptosporidium* infection on intestinal epithelial cells, as infection prevents induction of high levels of epithelial cell apoptosis early after infection when the parasite depends on the host cell for growth and development (McCole DF, et al. 2000, Infect Immun; Chen XM, et al., 2001, Gastroenterol; Liu J, et al., 2009, Infect Immun). Therefore, the level of parasite replication may not be correlated to the tissue damage in the intestine. Indeed, our RNA-Seq data demonstrate that the expression levels of most other inflammatory genes (except type I IFN-controlled genes) and genes that code proteins for cell proliferation and apoptosis were comparable between infected *Villin.Ifnar*^{1-/-} and *Ifnar*^{1^{fl/fl}} animals, which we believe may account for the similar morphological features and intestinal damage observed between infected *Villin.Ifnar*^{1-/-} and *Ifnar*^{1^{fl/fl}} groups (at 48 and 72h p.i.). We have included this explanation in the revised manuscript (lines 175-176).

Responses to comments from Reviewer #2

The authors have made revisions to the manuscript in light of reviewers' comments. Their results corroborate with previously published data showing upregulation of type I IFNs, IFN-??-mediated signaling and cytokine-mediated signaling pathways. As cited by the authors (and reported by Gibson et al 2022 PLoS Pathogens), *Cryptosporidium* oocyst shedding was significantly reduced in the *Ifnar*^{1-/-} mice compared to wild type. Although they have cited previous studies, the authors did not fully discuss the novelty of their findings and impact of their work ----- We thank you for your suggestion and have added discussion on the novelty and impact of these previous studies in the revised manuscript (lines 70-72 and 79-83).

The authors have not fully highlighted and discussed the novelty or impact of their work. The role of interferons (type I/III and IFN-gamma) in *Cryptosporidium* infection has been previously reported. Most of their work replicate previously reported findings, with only an incremental increase in new knowledge ----- We have modified the sentences in different text sections to highlight the novelty and impact of this study, with a focus on the detrimental effects of type I IFN signaling in intestinal epithelial cells on intestinal anti-*C. parvum* defense and the activation of type I IFN signaling through delivery of CSpV1-dsRNAs (lines 36, 95, 348-349, 436-438). We believe that our study not only provides new insights to the role of IFN signaling to mucosal anti-*C. parvum* defense (i.e., type I IFN signaling in the intestinal epithelial cells is detrimental to intestinal immunity to *Cryptosporidium* infection) but also reveals a new strategy of immune evasion by the parasite (i.e., through host delivery of CSpV1-dsRNAs to induce type I IFN response in host cells), relevant to the future development of effective therapeutic strategies for this critically important pathogen.

There are discrepancies in the data analysis and interpretation. These include: there is a discrepancy between the replicates shown in the RNAseq data heatmaps in Figure 4 (Fig. 4C and 4E) ----- We apologize for these discrepancies in the data analysis and interpretation and have corrected these

discrepancies between the replicates shown in the RNAseq data heatmaps in Figure 2 and 4 (lines 142, 237-238, 244, and 246).

Details on RNAseq analysis is still lacking. Quality controls and cut-offs for assessing poor quality reads are not discussed. ----- We have included more details on RNAseq analysis in the Method section, including the quality controls and cut-offs (lines 579 and 582-598).

Poor quality images of immunofluorescence staining of intestinal epithelium. It is hard to see parasites in the representative image. Not clear how the authors managed to count parasites in these images. ----- That's a good point, most likely due to the small size of these images in general. To make it more clearly on the fluorescent staining of the parasites in intestinal epithelium, we have added images of higher magnifications of the boxed regions in Fig 1c, showing strong and clear staining of the parasites for counting (new higher magnification insets in Fig 1c).

The exact mechanism of action of nitazoxanide is not known, therefore hinging on its potential antiviral activity may be an over-interpretation of data. ----- We agree and have modified the interpretation in the revised text (lines 436-4387).

Responses to comments from Reviewer #3

Deng et al, reports that *C. parvum* induce type I IFN response by delivery of CSpV1- dsRNAs to its host. Also, author analyzed the activation of pathway PKR/ RIG/MAV/STING by CSpV1- dsRNAs. Finally, the authors study the effects of anti-viral reagents and anti-parasite drugs on *C. parvum* infection of intestinal epithelium. Deng et al have addressed all previous suggestions and the manuscript have improved considerably. There are seven aspects that still need to be improved ----- Thank you for your positive comments about our revisions. We have revised the manuscript text to address all your concerns as detailed below.

1. Fig 2b says *C. parvum* (48 h) when the legend and the manuscript say 24 h. Please correct and clarify. Fig 2b Legend say two-tail, it should be changed to two-tailed. Fig 2e shows the data at 24 hpi but the manuscript (line 161) and the legend of the figure says 48 hpi, please correct. ----- We have corrected these mistakes (lines 142, 244, 246, and figure legends).

2. Please re-write lines 146-153. It is not clear what you are comparing in Fig 2c, only uninfected mice? --- We have re-written these lines to clearly show what is compared for the data analysis, including the analysis in Fig 2c (lines 148-157).

3. Line 171. Please, cite the correct figure. ----- This mistake has been corrected (line 174).

4. Line 193. Please, improve sentence. ----- We have modified this sentence and thanks for your suggestion (lines 196-200).

5. It will be great to have an illustration like reference 37 or propose a model what is happening in *C. parvum* and their CSpV1-dsRNAs. ----- We have added such an illustration in Fig 5g (line 277 and Fig 5g legend).

6. In line 339, reference 17, please add that IFN type I response reduced oocyst shedding and symptoms but not intestinal parasite burden in IFN γ KO mice. ----- We have added this to the sentence (lines 348-

349).

7. In line 348. Please, explain why IFN type I displays pro-parasitic effects. ----- We have modified the sentence to include explanation why IFN type I displays pro-parasitic effects (lines 358-360).

8. Normally, the quality of reads is determined by (FASQC), a threshold is selected and only reads that meet the thresholds are used for further analysis. Also, the data normally is trimmed to remove low-quality reads (Trimmomatic). Please supply more details on the RNAseq- analysis. ----- We have included more details on RNAseq analysis in the Method section, including the quality controls and threshold selection, in the revised manuscript (lines 579 and 582-598).